# Complex Query Answering on Eventuality Knowledge Graph with Implicit Logical Constraints

**Jiaxin Bai**
Department of CSE
HKUST
jbai@connect.ust.hk

**Xin Liu**
Department of CSE
HKUST
xliucr@cse.ust.hk

**Weiqi Wang**
Department of CSE
HKUST
wwangbw@cse.ust.hk

**Chen Luo**
Amazon.com Inc
cheluo@amazon.com

**Yangqiu Song**[*]
Department of CSE
HKUST
yqsong@cse.ust.hk

## Abstract

Querying knowledge graphs (KGs) using deep learning approaches can naturally leverage the reasoning and generalization ability to learn to infer better answers. Traditional neural complex query answering (CQA) approaches mostly work on entity-centric KGs. However, in the real world, we also need to make logical inferences about events, states, and activities (i.e., eventualities or situations) to push learning systems from System I to System II, as proposed by Yoshua Bengio. Querying logically from an EVentuality-centric KG (EVKG) can naturally provide references to such kind of intuitive and logical inference. Thus, in this paper, we propose a new framework to leverage neural methods to answer complex logical queries based on an EVKG, which can satisfy not only traditional first-order logic constraints but also implicit logical constraints over eventualities concerning their occurrences and orders. For instance, if we know that *Food is bad* happens before *PersonX adds soy sauce*, then *PersonX adds soy sauce* is unlikely to be the cause of *Food is bad* due to implicit temporal constraint. To facilitate consistent reasoning on EVKGs, we propose Complex Eventuality Query Answering (CEQA), a more rigorous definition of CQA that considers the implicit logical constraints governing the temporal order and occurrence of eventualities. In this manner, we propose to leverage theorem provers for constructing benchmark datasets to ensure the answers satisfy implicit logical constraints. We also propose a Memory-Enhanced Query Encoding (MEQE) approach to significantly improve the performance of state-of-the-art neural query encoders on the CEQA task.

## 1 Introduction

Querying knowledge graphs (KGs) can support many real applications, such as fact-checking and question-answering. Using deep learning methods to answer logical queries over KGs can naturally leverage the inductive reasoning and generalization ability of learning methods to overcome the sparsity and incompleteness of existing KGs, and thus has attracted much attention recently, which are usually referred to as Complex Query Answering (CQA) [37, 26, 38]. As the computational complexity of answering complex logical queries increases exponentially with the length of the query [37, 26], brute force search and sub-graph matching algorithms [29, 30, 28] are unsuitable

---

[*]Prof. Yangqiu Song is a visiting academic scholar at Amazon.

| Queries | Type | Interpretations |
|---|---|---|
| $q_1 = V_?. \exists V : \text{Interact}(V_?, V)$ $\wedge \text{Assoc}(V, Alzheimer) \wedge \text{Assoc}(V, MadCow)$ | Entity | Find the substances that interact with the proteins associated with Alzheimer's and Mad cow disease. |
| $q_2 = V_?. \text{Precedence}(Food\ is\ bad, PersonX\ add\ soy\ sauce)$ $\wedge \text{Reason}(Food\ is\ bad, V_?)$ | Eventuality | Food is bad before PersonX add soy sauce. What is the reason for food being bad? |
| $q_3 = V_?. \text{Precedence}(V_?, PersonX\ go\ home)$ $\wedge \text{ChosenAlternative}(PersonX\ go\ home, PersonX\ buy\ an\ umbrella)$ | Eventuality | Instead of buying an umbrella, PersonX go home. What happened before PersonX go home? |

Figure 1: Complex query examples and corresponding interpretations in natural language. $q_1$ is a query on an entity knowledge graph, while $q_2$ and $q_3$ are queries on an eventuality knowledge graph.

for processing complex queries. To overcome these challenges, various techniques, such as query encoding [23] and query decomposition [2], have been proposed. These techniques enable effective and scalable reasoning on incomplete KGs and facilitate the processing of complex queries.

Most of the existing work in this field has primarily focused on entity-centric KGs that only describe entities and their relationships. As Yoshua Bengio described in his view[2] of moving from System I to System II [16–18, 25, 13], we need to equip machine learning systems with logical, sequential reasoning, and other abilities. Particularly, such a system requires the understanding of how actions (including events, activities, or processes) interact with changes in distribution which can be reflected by states. Here we can summarize events, activities, and states as a linguistic term, eventualities (or situations), according to the linguistics literature [33, 4]. As with many other KG querying tasks, querying eventuality-centric knowledge graphs can also support many applications, such as providing references for making logical and rational decisions of intuitive inferences or eventual planning. This requires the CQA models to perform reasoning at the eventuality level. To provide resources for achieving eventuality-level reasoning, recently constructed KGs, such as ATOMIC [41, 24], Knowlywood [43], and ASER [52, 53], tend to use one or more discourse relations to represent the relationships between eventuality instances. For example, *PersonX went to the store* and *PersonX bought some milk* are two simple eventuality instances, with the latter being a possible consequence of the former. The construction of these EVentuality-centric Knowledge Graphs (EVKGs) thoroughly maps the relationships between eventualities and enables us to reason about eventuality instances and their relationships using logical queries, thereby facilitating a more comprehensive approach to modeling complex relationships than traditional knowledge graphs.

Aside from the importance of querying EVKGs, reasoning on EVKG also significantly differs from that on an entity-centric KG because eventualities involve considering their occurrences and order. In entity-centric KGs, as shown in Figure 1 $q_1$, the vertices represent entities such as *Alzheimer* or *Mad Cow Disease*, and truth values are assigned to the edges between entities to indicate their relationships. For example, the statement $\text{Assoc}(Beta-amyloid, Alzheimer)$ is true. In contrast, during the reasoning process on EVKG, the eventualities may or may not occur, and determining their occurrence is a crucial part of the reasoning. For instance, given $\text{ChosenAlternative}(PersonX\ go\ home, PersonX\ buy\ umbrella)$ in Figure 1 $q_2$, it implicitly suggests that "PersonX go home" occurs, while "PersonX buy umbrella" does not. Moreover, there are relationships that explicitly or implicitly describe the order of occurrences, such as temporal and causal relations. For example, $\text{Reason}(PersonX\ study\ hard, PersonX\ pass\ exam)$ indicates the causality between "PersonX pass the exam" and "PersonX study hard," which also implies that "PersonX pass the exam" occurs after "PersonX study hard." When multiple edges are presented in a given situation, it is essential to ensure that there are no contradictions regarding the occurrence of these eventualities. For example, in Figure 1 $q_3$, $\text{ChosenAlternative}(PersonX\ go\ home, PersonX\ buy\ umbrella) \wedge \text{Succession}(PersonX\ go\ home, PersonX\ buy\ umbrella)$ is contradictory because the former suggests that PersonX did not buy an umbrella, while the latter implies otherwise.

To enable complex reasoning on eventuality knowledge graphs, we formally define the problem of complex eventuality query answering (CEQA). CEQA is a more rigorous definition of CQA on EVKG that consider not only the explicitly given relational constraints, but also the implicit logical constraints on the occurrences and temporal order of eventualities. The implicit constraints are derived from the relational constraints and can be further divided into two types: *occurrence constraints* and *temporal constraints*. Incorporating these implicit constraints into complex query answers drastically

---

[2] http://www.iro.umontreal.ca/~bengioy/AAAI-9feb2020.pdf

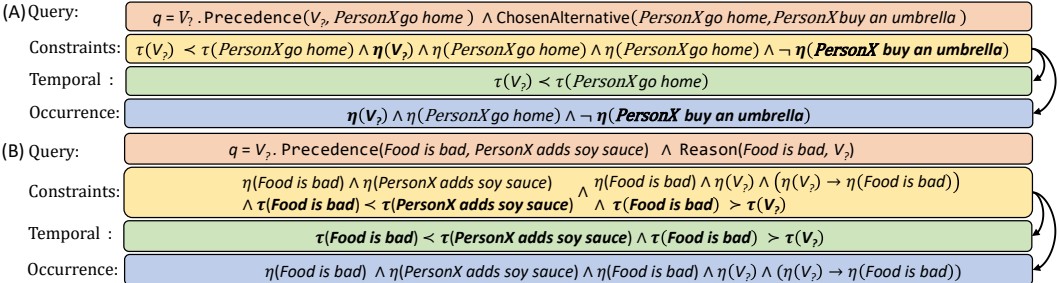

Figure 2: Complex eventuality queries with their implicit temporal and occurrence constraints

changes the nature of the reasoning process. Unlike conventional CQA, the reasoning process of CEQA is defeasible because when additional knowledge is presented, the original reasoning could be weakened and overturned [19]. For example, we showed in Figure 2, *PersonX adds soy sauce* is a possible answer to the query *What is the reason for food being bad*. However, if more knowledge is given, like *Food is bad* is before *PersonX adds soy sauce*, then it cannot be the proper reason anymore due to temporal constraints. However, all the existing methods for CQA cannot incorporate additional knowledge to conduct defeasible reasoning in CEQA.

To address this problem, we propose the method of memory-enhanced query encoding (MEQE). In the MEQE method, we first separate the logic terms in a query into two categories, computational atomics and informational atomics. Computational atomics, like $\texttt{Reason}(Food\,is\,bad, V_?)$, contains at least one variable in their arguments, and informational atomics, like $\texttt{Precedence}(Food\,is\,bad, PersonX\,add\,soy\,sauce)$, does not contain variables. For the computational atomics, following previous work, we construct the corresponding computational graph to recursively compute its query embedding step-by-step. For the informational atomics, we put them into a key-value memory module. For each of the informational atomics, its head argument is used as the memory key, and its relation type and tail arguments are used as memory values. In the query encoding process, after each operation in the computational graph, a relevance score is computed between the query embedding and memory heads. This relevance score is then used to retrieve the corresponding memory values of the corresponding relation and tail. Then these memory values are aggregated, adjusted, and added back to the query embedding. By doing this, the query encoder is able to leverage implicit logical constraints that are given by the informational atomics. We evaluate our proposed MEQE method on the eventuality knowledge graph, ASER, which involves fourteens types of discourse relations between eventualities. Experiment results show that our proposed MEQE is able to consistently improve the performance of four frequently used neural query encoders on the task of CEQA. Code and data are publicly available [3].

## 2 Problem Definition

In this section, we first introduce the definitions of the complex queries on entity-centric and eventuality-centric KGs. Then we give the definition of implicit logical constraints and the informational atomics that specifically provide such constraints to the eventuality queries.

### 2.1 Complex Queries

Complex query answering is conducted on a KG: $\mathcal{G} = (\mathcal{V}, \mathcal{R})$. The $\mathcal{V}$ is the set of vertices $v$, and the $\mathcal{R}$ is the set of relation $r$. The relations are defined in functional forms to describe the logical expressions better. Each relation $r$ is defined as a function with two arguments representing two vertices, $v$ and $v'$. The value of function $r(v, v') = 1$ if and only if there is a relation between the vertices $v$ and $v'$.

In this paper, the queries are defined in conjunctive forms. In such a query, there are logical operations such as existential quantifiers $\exists$ and conjunctions $\wedge$. Meanwhile, there are anchor eventualities $V_a \in \mathcal{V}$, existential quantified variables $V_1, V_2, ...V_k \in \mathcal{V}$, and a target variable $V_? \in \mathcal{V}$. The query

---

[3] https://github.com/HKUST-KnowComp/CEQA

Table 1: The discourse relations and their implicit logical constraints. $\eta(V)$ is `True` if and only if $V$ occurs. $\tau(V)$ indicates the happening timestamp of $V$. Meanwhile, the instance-based temporal logic operator $\prec$, $\succ$, or $=$ means $V_1$ is before, after, or at the same time as $V_2$.

| Discourse Relations ($e_i$) | Semantics | Implicit Constraints | |
|---|---|---|---|
| | | Occurrence Constraints ($o_i$) | Temoral Constraints ($t_i$) |
| `Precedence`$(V_1, V_2)$ | $V_1$ occurs before $V_2$. | $\eta(V_1) \wedge \eta(V_2)$ | $\tau(V_1) \prec \tau(V_2)$ |
| `Succession`$(V_1, V_2)$ | $V_1$ occurs after $V_2$ happens. | $\eta(V_1) \wedge \eta(V_2)$ | $\tau(V_1) \succ \tau(V_2)$ |
| `Synchronous`$(V_1, V_2)$ | $V_1$ occurs at the same time as $V_2$. | $\eta(V_1) \wedge \eta(V_2)$ | $\tau(V_1) = \tau(V_2)$ |
| `Reason`$(V_1, V_2)$ | $V_1$ occurs because $V_2$. | $\eta(V_1) \wedge \eta(V_2) \wedge (\eta(V_1) \leftarrow \eta(V_2))$ | $\tau(V_1) \succ \tau(V_2)$ |
| `Result`$(V_1, V_2)$ | $V_1$ occurs as a result $V_2$. | $\eta(V_1) \wedge \eta(V_2) \wedge (\eta(V_1) \rightarrow \eta(V_2))$ | $\tau(V_1) \prec \tau(V_2)$ |
| `Condition`$(V_1, V_2)$ | If $V_2$ occurs, $V_1$. | $\eta(V_1) \rightarrow \eta(V_2)$ | $\tau(V_1) \succ \tau(V_2)$ |
| `Concession`$(V_1, V_2)$ | $V_2$ occurs, although $V_1$. | $\eta(V_1) \wedge \eta(V_2)$ | - |
| `Constrast`$(V_1, V_2)$ | $V_2$ occurs, but $V_1$. | $\eta(V_1) \wedge \eta(V_2)$ | - |
| `Conjunction`$(V_1, V_2)$ | $V_1$ and $V_2$ both occur. | $\eta(V_1) \wedge \eta(V_2)$ | - |
| `Instantiation`$(V_1, V_2)$ | $V_2$ is a more detailed description of $V_1$. | $\eta(V_1) \wedge \eta(V_2)$ | - |
| `Restatement`$(V_1, V_2)$ | $V_1$ restates the semantics of $V_2$. | $\eta(V_1) \leftrightarrow \eta(V_2)$ | - |
| `Alternative`$(V_1, V_2)$ | $V_1$ and $V_2$ are alternative situations. | $\eta(V_1) \vee \eta(V_2)$ | - |
| `ChosenAlternative`$(V_1, V_2)$ | Instead of $V_2$ occurs, $V_1$. | $\eta(V_1) \wedge \neg\eta(V_2)$ | - |
| `Exception`$(V_1, V_2)$ | $V_1$, except $V_2$. | $\neg\eta(V_1) \wedge \eta(V_2) \wedge (\neg\eta(V_2) \rightarrow \eta(V_1))$ | - |

is written to find the answers $V_? \in \mathcal{V}$, such that there exist $V_1, V_2, ... V_k \in \mathcal{V}$ satisfying the logical expression:

$$q[V_?] = V_?.\exists V_1, ..., V_k := e_1 \wedge e_2 \wedge ... \wedge e_m. \tag{1}$$

Each $e_i$ is an atomic expression in any of the following forms: $e_i = r(v_a, V)$, or $e_i = r(V, V')$. Here $v_a$ is an anchor eventuality, and $V, V' \in \{V_1, V_2, ..., V_k, V_?\}$ are distinct variables.

## 2.2 Complex Eventuality Queries

For complex eventuality queries, they can also be written in the form of a conjunctive logical expression as Eq. (1). Differently, each atomic $e_i$ can all be in the form of $e_i = r(v_i, v_j)$, where $v_i, v_j \in V$ are given eventualities. These atomics, which do not include variables, are called informational atomics, because they only provide implicit constraints .

The relations $r$ in CEQA are discourse relations, and they exert implicit constraints over the eventualities, and these constraints can be categorized into occurrence constraints and temporal constraints. Suppose the occurrence and temporal constraints derived from the $i$-th atomic $e_i$ is denoted as $o_i$ and $t_i$. Then complex eventuality query, including its implicit constraints can be written as

$$q[V_?] = V_?.\exists V_1, ..., V_k := (e_1 \wedge ... \wedge e_m) \wedge (o_1 \wedge ... \wedge o_m) \wedge (t_1 \wedge ... \wedge t_m). \tag{2}$$

The constraints brought from each type of discourse relations are presented in Table 1. Further justifications of the derivation process are given in the Appendix B.

### 2.2.1 Occurrence Constraints

The occurrence constraints determine whether certain eventuality occurs or not. For instance, consider Figure 2 (A), where the logical query means that *Instead of buying an umbrella, PersonX goes home. What occurred before PersonX went home?* If we rely solely on relational constraints, as in the conventional definition of CQA, the answers are only determined by the latter part of the query, *What happened before PersonX went home?* Consequently, *PersonX buys an umbrella* could be a solution to this query. However, within the query, there is an information atomic saying, *instead of buying an umbrella, PersonX goes home*, which denies the occurrence of *PersonX buying an umbrella.* To formally express such constraint, we use the function $\eta(V)$. If eventuality $V$ occurs, then $\eta(V) =$ `True`, otherwise it is `False`. As depicted in Figure 2, the occurrence constraint of this query comprises the terms $\eta(V_?) \wedge \neg\eta(PersonX\,buy\,umbrella)$. In this case, $V_?$ cannot be *PersonX buys an umbrella* or there is an contradiction.

Most discourse relations assume the occurrence of the argument eventualities, for example, `Precedence`, `Conjunction`, and `Reason`. However, there are also relations that do not imply the occurrence of the arguments, such as `Condition` and `Restatement`. Moreover, the `Exception` and `ChosenAlternative` relations restrict certain eventualities from happening. For instance, in the

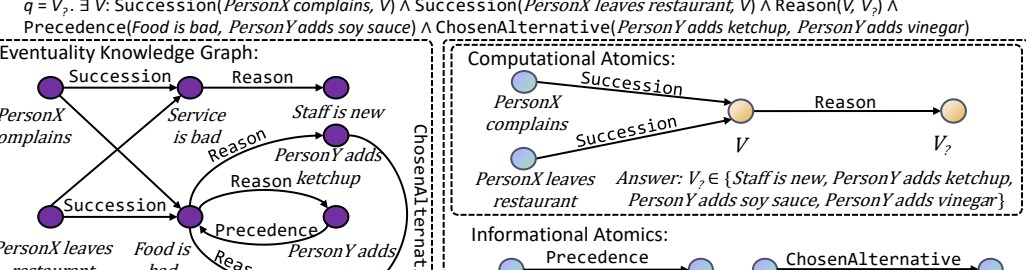

Figure 3: An example complex eventuality query with the computational and informational atomics. $V$ is something that happens before a person complains and leaves the restaurant, according to the KG, the $V$ could be either *Service is bad* or *Food is bad*. If $V_?$ is the reason of $V$, then according to the graph, $V_?$ could be either *Staff is new*, *PersonY adds ketchup*, *PersonY adds soy sauce*, and *PersonY adds vinegar*. However, in the query we also know that *PersonY adds vinegar* does not happen, and *PersonY adds soy sauce* happens after the *Food is bad*, thus cannot be the reason for *Food is bad*. The conflict here is causality implies precedence.

case of `ChosenAlternative`$(PersonX\,read\,books, PersonX\,play\,games)$, it implies that PersonX reads books: $\eta(PersonX\,read\,books)$, and does not play games: $\neg\eta(PersonX\,play\,games)$. Another example is `Exception`$(Room\,is\,empty, PersonX\,stay\,in\,room)$, which implies that the room is not empty and PersonX is present in the room. Furthermore, if PersonX is not in the room, then the room is empty. This can be formally expressed as $\neg\eta(Room\,is\,empty) \wedge \eta(PersonX\,stay\,in\,room) \wedge (\neg\eta(PersonX\,stay\,in\,room) \rightarrow \eta(Room\,is\,empty))$. For a comprehensive overview of the occurrence constraints, please refer to Table 1.

### 2.2.2 Temporal Constraints

The temporal constraints reflect the order of occurrence of the eventualities. As shown in Figure 2 (B), the complex query on the eventuality knowledge graph can be interpreted as *Food is bad before PersonX adds soy sauce. What is the reason for food being bad?* If we only considered the relational constraints, like in the conventional setting of CQA, then *PersonX adds soy sauce* is a possible answer. However, in the definition of CEQA, the answer *PersonX adds soy sauce* is incorrect because the food is bad already occurred before *PersonX added soy sauce*, but something that occurs later is impossible to be the reason for something that previously occurred. Formally, we use the expression of temporal logic $\succ$, $\prec$, and $=$ to describe the temporal order between two eventualities [22]. $\tau(A) \prec \tau(B)$ means $A$ occurs before $B$, and $\tau(A) = \tau(B)$ means they happen at the same time, and $\tau(A) \succ \tau(B)$ means $A$ occurs after $B$. For example in Figure 2 (B), the temporal constraint is represented by $\tau(Food\,is\,bad) \prec \tau(PersonX\,add\,soy\,sauce) \wedge \tau(Food\,is\,bad) \succ \tau(V_?)$, which can be interpreted as *Food is bad* is before *PersonX adds soy sauce* and $V_?$ is before *Food is bad*. Because of this, $V_?$ cannot *PersonX adds soy sauce*, otherwise there exists a contradiction.

The temporal relations `Precedence`$(A, B)$, `Succession`$(A, B)$, and `Synchronous`$(A, B)$ naturally describes the temporal constraint. Meanwhile, previous studies also assume that causation implies precedence [40, 10, 54], With this assumption, the temporal constraints can also be derived from relations like `Reason` and `Result`. The descriptions of temporal constraints are given in Table 1.

## 3 Memory-Enhanced Query Encoding

In this section, we first introduce the method of query encoding, and then introduce how to use the memory module to represent the informational atomics to conduct reasoning on EVKGs.

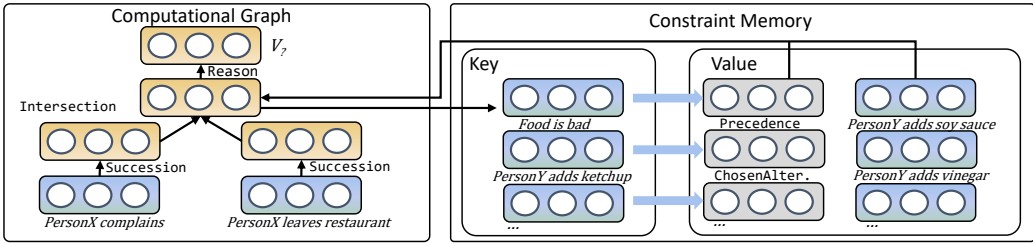

Figure 4: The example computational graph and the memory-enhanced query encoding process.

## 3.1 Computational Graph and Query Encoding

Figure 3 and 4 show that there is a computational graph for each query. This computational graph is a directed acyclic graph (DAG) that consists of nodes and edges representing intermediate encoding states and neural operations, respectively. By recursively encoding the sub-queries following the computational graph, the operations implicitly model the set operations of the intermediate query results. The set operations are defined as follows: (1) *Relational Projection*: Given a set of vertices $A$ and a relation $r \in R$, the relational projection operation returns all eventualities that hold the relation $r$ with at least one entity $e \in A$. This can be expressed as: $P_r(A) = \{v \in \mathcal{V} \mid \exists v' \in A, r(v', v) = 1\}$; (2) *Intersection*: Given sets of eventualities $A_1, \ldots, A_n \subseteq \mathcal{V}$, the intersection computes the set that is the subset to all of the sets $A_1, \ldots, A_n$. This can be expressed as $\bigcap_{i=1}^{n} A_i$.

Various query encoding methods are proposed to recursively encode the computational graph. However, the query embeddings of these methods can be translated into $d$-dimensional vectors. As shown in Figure 4, the computations along the computation graph start with the anchor eventualities, such as *PersonX complains*. Suppose the embedding of an anchor $v$ is denoted as $e_v \in R^d$. Then, the initial query embedding is computed as $q_0 = e_v$. As for the *relational projection* operation, suppose the $e_{rel} \in R^d$ is the embedding vector of the relation $rel$. The relation projection $F_{proj}$ is expressed as

$$q_{i+1} = F_{proj}(q_i, e_{rel}), \tag{3}$$

where the $q_i$ and $q_{i+1}$ are input and output query embeddings for this relational projection operation.

Meanwhile, for the *Intersection* operations, suppose there are $k$ embeddings of sub-queries, $q_i^{(1)}, q_i^{(2)}, ..., q_i^{(k)}$, as the input for this operation, then the output can be expressed as:

$$q_{i+1} = F_{inter}(q_i^{(1)}, q_i^{(2)}, ..., q_i^{(k)}), \tag{4}$$

where the $F_{inter}$ is a neural network that is permutation-invariant to the input sub-query embeddings adopted from the backbone models [23, 5, 1, 12].

## 3.2 Memory-Enhanced Query Encoding

The computational graph is capable of encoding computational atomics presented in the logical expression. However, informational atomics can influence the reasoning outcomes by introducing implicit temporal or occurrence constraints. As depicted in Figure 3, the absence of informational atomics results in two false answers from the knowledge graph. When informational atomics are included, providing implicit constraints, the only two correct answers can be derived.

Based on this observation, we propose using a memory module to encode the constraint information provided by the informational atomics. Suppose that there are $M$ informational atomics in the query. Their head embeddings, relation embeddings, and tail embeddings are represented as $c_h^{(m)}, c_r^{(m)}$, and $c_t^{(m)}$ respectively. For each operator output $q_i$ from the computational graph, we compute its relevance score $s_{i,m}$ towards each head eventuality $m$,

$$s_{i,m} = <q_i, c_h^{(m)}> . \tag{5}$$

Then we use the $s_{i,m}$ to access the values from the constraint relation and tails, and then aggregate the memory values according to the relevance scores

$$v_i = \sum_{m=1}^{M} s_{i,m}(c_r^{(m)} + c_t^{(m)}). \tag{6}$$

Table 2: The dataset details for CEQA. #Ans. reports the number of answers that are proved to be not contradictory by theorem provers. #Contr. Ans. reports the number of answers that can be searched from the ground truth KG, but are contradictory due to the occurrence or temporal constraints.

| Data Split | #Types | OccurrenceConstarints | | | Temporal Constraints | | |
|---|---|---|---|---|---|---|---|
| | | #Queries | #Ans. | #Contr. Ans. | #Queries | #Ans. | # Contr. Ans. |
| Train | 6 | 124,766 | 5.02 | 1.53 | 35,962 | 5.02 | 1.15 |
| Validation | 15 | 30,272 | 7.68 | 1.75 | 23,905 | 9.17 | 1.44 |
| Test | 15 | 30,243 | 8.40 | 1.81 | 24,226 | 11.40 | 1.50 |

Finally, as shown in Figure 4, the constraint values are added back to the query embedding after going through a feed-forward layer FFN, and this process is described by

$$q_i = q_i + \text{FFN}(v_i). \tag{7}$$

## 3.3 Learning Memory-Enhanced Query Encoding

To train the model, we compute the normalized probability of $v$ being the correct answer to query $q$ by applying the softmax function to all similarity scores:

$$p(q, v) = \frac{e^{<q, e_v>}}{\sum_{v' \in V} e^{<q, e_{v'}>}}, \tag{8}$$

where $< \cdot, \cdot >$ denotes the dot product of two vectors, when $q$ is the query embedding after the last operation. A cross-entropy loss is used to maximize the log probabilities of all correct answer pairs:

$$\mathcal{L} = -\frac{1}{N} \sum_{i=1}^{N} \log p(q^{(i)}, v^{(i)}), \tag{9}$$

where $(q^{(i)}, v^{(i)})$ denotes one of the positive query-answer pairs, and $N$ is the total number of them.

## 4 Experiments

To ensure a fair comparison of various methods for the CEQA problem, we generated a dataset by sampling from ASER [53], the largest eventuality knowledge graph, which encompasses fourteen types of discourse relations. The division of edges within each knowledge graph into training, validation, and testing sets was performed in an 8:1:1 ratio, as illustrated in Table 5. The training graph $\mathcal{G}_{train}$, validation graph $\mathcal{G}_{val}$, and test graph $\mathcal{G}_{test}$ were constructed using the training edges, training+validation edges, and training+validation+testing edges, respectively, following the established configuration outlined in prior research by [37]. Moreover, we conducted evaluations using different reasoning models, consistent with settings in previous studies.

### 4.1 Query Sampling with Theorem Prover

We employ the sampling algorithm proposed by [37] with the conjunctive query types outlined in [46]. Specifically, for the training dataset, we sample queries that have a maximum of two anchor nodes, while for the validation and test sets, we select queries containing up to three anchor eventualities. The query types in our framework reflect the structure of the computational graph and are represented using a Lisp-like format [46, 7]. Once the query-answer pairs are sampled, we randomly select up to three edges that share common vertices with the reasoning chain of the query-answer pairs. These selected edges are then used as the informational atomics for the corresponding query. Subsequently, we employ the z3 prover [15] to filter the queries. We retain only those queries where the informational atomics incorporate effective implicit constraints, ensuring the presence of meaningful constraints in the data. The detailed query types and their numbers of answer with/without contradictions are shown in Table 6, in which the p is for projection, the i is for intersection, and e is for eventuality.

In detail, for each eventuality present on the reasoning path towards an answer in the complex query, we create a corresponding boolean variable in the z3 prover. We then incorporate the relevant

Table 3: Experiment results of different query encoding models. In this experiment, we compare the performance of the query encoder with or without the memory-enhanced query encoding method.

| Models | OccurrenceConstraints | | | Temporal Constraints | | | Average | | |
| --- | --- | --- | --- | --- | --- | --- | --- | --- | --- |
| | Hit@1 | Hit@3 | MRR | Hit@1 | Hit@3 | MRR | Hit@1 | Hit@3 | MRR |
| GQE | 8.92 | 14.21 | 13.09 | 9.09 | 14.03 | 12.94 | 9.12 | 14.12 | 13.02 |
| + MEQE | **10.20** | **15.54** | **14.31** | **10.70** | **15.67** | **14.50** | **10.45** | **15.60** | **14.41** |
| Q2P | 14.14 | 19.97 | 18.84 | 14.48 | 19.69 | 18.68 | 14.31 | 19.83 | 18.76 |
| + MEQE | **15.15** | **20.67** | **19.38** | **16.06** | **20.82** | **19.74** | **15.61** | **20.74** | **19.56** |
| Nerual MLP | 13.03 | 19.21 | 17.75 | 13.45 | 19.06 | 17.68 | 13.24 | 19.14 | 17.71 |
| + MEQE | **15.26** | **20.69** | **19.32** | **15.91** | **20.63** | **19.47** | **15.58** | **20.66** | **19.40** |
| FuzzQE | 11.68 | 18.64 | 17.07 | 11.68 | 17.97 | 16.53 | 11.68 | 18.31 | 16.80 |
| + MEQE | **14.76** | **21.12** | **19.45** | **15.31** | **21.01** | **19.49** | **15.03** | **21.06** | **19.47** |

occurrence constraints based on the relations between these eventualities, as outlined in Table 1, and feed them into the z3 prover. If the result returned by the prover is `unsat`, it indicates a contradiction in the reasoning process. Regarding temporal constraints, we follow a similar approach. We create corresponding floating variables that represent the timestamps of the occurrence of the eventualities. We then establish constraints on the temporal order by utilizing floating operators such as >, =, or < between the variables. By doing so, for each query, we establish a corresponding linear program. Once again, if the prover outputs `unsat`, it signifies a contradiction, namely, there is no solution for the timestamps of these events. Queries that have no contradictory answers and queries where all the answers are contradictory are discarded. The remaining queries are then categorized into two types: queries with occurrence constraints and queries with temporal constraints. Table 6 presents the average number of contradictory and non-contradictory answers per query.

## 4.2 Baselines and Metrics

In this section, we introduce several baseline query encoding models that use different neural network architectures to parameterize the operators in the computational graph and recursively encode the query into various embedding structures: (1) GQE [23] uses vectors to encode complex queries; (2) Q2P [5] uses multiple vectors to encode queries; (3) Neural MLP [1] use MLP as the operators; (4) FuzzQE [12] uses fuzzy logic to represent logical operators.

To define the evaluation metrics, we use $q$ to represent a testing query, and $\mathcal{G}_{val}$ and $\mathcal{G}_{test}$ to represent the validation and testing knowledge graphs, respectively. We use $[q]_{val}$ and $[q]_{test}$ to represent the answers to query $q$ on $\mathcal{G}_{val}$ and $\mathcal{G}_{test}$, respectively. Eq. (10) shows how to compute the metrics. When the evaluation metric is Hit@K, $m(r)$ is defined as $m(r) = \mathbf{1}[r \leq K]$, where $m(r) = 1$ if $r \leq K$, and $m(r) = 0$ otherwise. For mean reciprocal ranking (MRR), $m(r)$ is defined as $m(r) = \frac{1}{r}$.

$$\texttt{metric}(q) = \frac{\sum_{v \in [q]_{test}/[q]_{val}} m(\texttt{rank}(v))}{|[q]_{test}/[q]_{val}|}.  \tag{10}$$

During the training process, the testing graph $\mathcal{G}_{test}$ is unobserved. In the hyper-parameters selection process, we use the same metrics as Eq. (10), but replace the graphs $\mathcal{G}_{test}/\mathcal{G}_{val}$ with $\mathcal{G}_{val}/\mathcal{G}_{train}$.

## 4.3 Details

To ensure fair comparisons, we replicate all the models under a unified framework. We use the same number of embedding sizes of three hundred for all models and use grid-search to tune the hyperparameters of the learning rate ranging from $\{0.002, 0.001, 0.0005, 0.0002, 0.0001\}$ and batch size ranging from $\{128, 256, 512\}$. All the experiments can be run on NVIDIA RTX3090 GPUs. Experiments are repeated three times, and the averaged results are reported.

Table 4: The Hit@3 and MRR on different query types with a various number of anchor nodes.

| #Anc. | Query Type | Metric | GQE | | Q2P | | Neural MLP | | FuzzQE | |
|---|---|---|---|---|---|---|---|---|---|---|
| | | | Base. | MEQE | Base. | MEQE | Base. | MEQE | Base. | MEQE |
| 2 | (p,(i,(p,(e)),(p,(e)))) | Hit@3 | 12.97 | **13.76** | 17.74 | **18.88** | 15.93 | **17.32** | 15.23 | **18.02** |
| | | MRR | 11.86 | **12.75** | 16.90 | **18.35** | 15.31 | **16.51** | 14.38 | **16.58** |
| | (i,(p,(e)),(p,(e))) | Hit@3 | 33.52 | **34.48** | **44.65** | 39.54 | 38.39 | **40.29** | **43.71** | 39.77 |
| | | MRR | 30.53 | **32.80** | **39.79** | 34.77 | 35.02 | **35.16** | **36.92** | 36.53 |
| | (i,(p,(e)),(p,(p,(e)))) | Hit@3 | 12.40 | **12.42** | 15.22 | **15.96** | 15.03 | **15.69** | 15.56 | **16.45** |
| | | MRR | **11.46** | 11.38 | 14.36 | **15.25** | 14.21 | **14.74** | 14.82 | **15.36** |
| | (i,(p,(p,(e))),(p,(p,(e)))) | Hit@3 | 14.16 | **14.87** | 17.49 | **19.86** | 17.06 | **19.07** | 16.58 | **18.65** |
| | | MRR | 13.16 | **13.19** | 16.48 | **18.89** | 15.49 | **18.27** | 14.69 | **17.22** |
| 3 | (p,(i,(i,(p,(e)),(p,(e))),(p,(e)))) | Hit@3 | 14.63 | **18.02** | 25.67 | **26.17** | 23.93 | **24.34** | 18.58 | **26.31** |
| | | MRR | 13.47 | **16.95** | 24.38 | **25.13** | 22.63 | **23.41** | 17.72 | **24.92** |
| | (i,(p,(e)),(p,(i,(p,(e)),(p,(e))))) | Hit@3 | 17.20 | **20.63** | 22.52 | **22.92** | 23.22 | **23.99** | 22.67 | **24.53** |
| | | MRR | 15.63 | **19.61** | 21.76 | **21.93** | 21.73 | **22.67** | 21.51 | **23.01** |
| | (i,(i,(p,(e)),(p,(e))),(p,(e))) | Hit@3 | 24.66 | **28.11** | **45.10** | 44.12 | 40.28 | **40.62** | 47.14 | **47.56** |
| | | MRR | 22.57 | **24.22** | **40.14** | 37.87 | 35.71 | **36.70** | 40.95 | **41.65** |
| | (i,(i,(p,(e)),(p,(p,(e)))),(p,(e))) | Hit@3 | 13.17 | **13.31** | **17.06** | 16.72 | 18.04 | **18.80** | 16.62 | **18.31** |
| | | MRR | 11.81 | **12.38** | **17.00** | 16.44 | 16.86 | **17.42** | 15.88 | **17.24** |
| | (i,(i,(p,(p,(e))),(p,(p,(e)))),(p,(e))) | Hit@3 | 16.94 | **19.63** | 22.06 | **22.94** | 21.66 | **23.85** | 19.70 | **22.65** |
| | | MRR | 15.62 | **17.59** | 20.76 | **21.60** | 20.45 | **22.19** | 17.52 | **21.70** |
| | (i,(p,(i,(p,(e)),(p,(e)))),(p,(p,(e)))) | Hit@3 | 16.23 | **19.75** | 24.45 | **25.59** | 23.39 | **25.45** | 22.33 | **25.63** |
| | | MRR | 15.05 | **18.36** | 23.30 | **24.15** | 21.60 | **24.26** | 20.87 | **24.00** |
| | (i,(i,(p,(e)),(p,(e))),(p,(p,(e)))) | Hit@3 | 20.43 | **23.08** | 34.52 | **36.44** | 36.56 | **42.00** | 35.88 | **41.80** |
| | | MRR | 19.26 | **21.74** | 31.91 | **33.45** | 32.46 | **37.41** | 33.74 | **36.65** |
| | (i,(i,(p,(e)),(p,(p,(e)))),(p,(p,(e)))) | Hit@3 | 13.29 | **15.05** | 20.08 | **20.87** | 21.79 | **22.94** | 19.65 | **22.81** |
| | | MRR | 12.34 | **14.04** | 19.31 | **19.81** | 19.57 | **21.65** | 17.85 | **21.37** |
| | (i,(i,(p,(p,(e))),(p,(p,(e)))),(p,(p,(e)))) | Hit@3 | 15.64 | **17.67** | 22.63 | **25.10** | 22.97 | **24.50** | 20.22 | **25.44** |
| | | MRR | 14.54 | **16.39** | 21.08 | **23.13** | 20.70 | **23.04** | 21.93 | **23.22** |

## 4.4 Experiment Results

Table 3 presents the results of the main experiment, which compares different query encoding models with and without MEQE. The table includes the performance metrics of Hit@1, Hit@3, and MRR for both occurrence constraints and temporal constraints, along with the average scores across all categories. The experimental results demonstrate that our proposed memory-enhanced query encoding (MEQE) model consistently improves the performance of existing query encoders in complex eventuality query answering. We conduct experiments on four commonly used query encoders, and the MEQE model, leveraging the memory model depicted in Figure 4, outperforms the baselines. The MEQE models differ structurally from the baseline models by incorporating a memory module that contains informational atomics. By reading this memory module, MEQE effectively incorporates implicit constraints from these atomics, leading to improved performance.

Additionally, we observed that combining MEQE with the Q2P [5] model yields the best average performance across three metrics: Hit@1, Hit@3, and MRR. Furthermore, on average, MEQE enhances the Hit@1 metric by 17.53% and the Hit@3 metric by 9.53%. The greater improvement in the Hit@1 metric suggests that the model's ability to accurately predict the top-ranked answer has improved more significantly compared to predicting answers within the top three rankings. Moreover, MEQE demonstrates a 13.85% improvement in performance on queries with temporal constraints and an 11.15% improvement on occurrence constraints. This indicates that MEQE is particularly effective in handling temporal constraints compared to occurrence constraints.

Table 4 displays the Hit@3 and MRR results of various types of complex queries. The table demonstrates the superiority of MEQE over the baseline models across different query types. Furthermore, the table indicates that, on average, MEQE achieves an improvement of 8.1% and 11.6% respectively. This suggests that MEQE is particularly adept at handling queries with multiple eventualities.

## 5 Related Work

Complex query answering is a task in deductive knowledge graph reasoning, where a system or model is required to answer a logical query on an incomplete knowledge graph. Query encoding [23] is a fast and robust method for addressing complex query answering. Various query embedding methods

utilize different structures to encode logical KG queries, enabling them to handle different types of logical queries. The GQE method, introduced by Hamilton et al. [23], represents queries as vector representations to answer conjunctive queries. Ren et al. [37] employed hyper-rectangles to encode and answer existential positive first-order (EPFO) queries. Simultaneously, Sun et al. [42] proposed the use of centroid-sketch representations to enhance the faithfulness of the query embedding method for EPFO queries. Both conjunctive queries and EPFO queries are subsets of first-order logic (FOL) queries. The Beta Embedding [36] is the first query embedding method that supports a comprehensive set of operations in FOL by encoding entities and queries into probabilistic Beta distributions. Moreover, Zhang et al. [55] utilized cone embeddings to encode FOL queries. Meanwhile, there are also neural-symbolic methods for query encoding. Xu et al. [49] proposes an entangled neural-symbolic method, ENeSy, for query encoding. Wang et al. [47] propose using pre-trained knowledge graph embeddings and one-hop message passing to conduct complex query answering. Additionally, Yang et al. [50] propose using Gamma Embeddings to encode complex logical queries. Finally, Liu et al. [27] propose pre-training on the knowledge graph with kg-transformer and then fine-tuning on the complex query answering. Recently, Bai et al. [7] proposes to use sequence encoders to encode the linearized computational graph of complex queries. Galkin et al. [20] propose to conduct inductive logical reasoning on KG, and Zhu et al. [56] proposes GNN-QE to conduct reasoning on KG with message passing on the knowledge graph. Meanwhile, Bai et al. [6] formulate the problem of numerical CQA and propose the corresponding query encoding method of NRN.

Another approach to addressing complex knowledge graph queries is query decomposition [2]. In this research direction, the probabilities of these atomic queries are modeled using link predictors, and then an inference time optimization is used to find the answers. In addition, an alternative to query encoding and query decomposition is proposed by Wang et al. [47]. They employ message passing on one-hop atomic queries to perform complex query answering. A recent neural search-based method called QTO is introduced by Bai et al. [8], which has shown impressive performance in complex question answering (CQA). Theorem proving is another deductive reasoning task applied to knowledge graphs. Neural theorem proving methods [39, 31, 32] have been proposed to tackle the incompleteness of KGs by using embeddings to conduct inference on missing information.

## 6 Limitation

Although our experiments demonstrate that MEQE improves the performance of existing models on the CEQA task, the evaluation is conducted on specific benchmark datasets constructed with theorem provers from the largest general-domain eventuality graph ASER [53]. The generalizability of the proposed approach to specific or professional fields may require further investigation and evaluation.

## 7 Conclusion

In this paper, we introduced complex eventuality query answering (CEQA) as a more rigorous definition of complex query answering (CQA) for eventuality knowledge graphs (EVKGs). We addressed the issue of implicit logical constraints on the occurrence and temporal order of eventualities, which had not been adequately considered in the existing definition of CQA. To ensure consistent reasoning, we leveraged theorem provers to construct benchmark datasets that enforce implicit logical constraints on the answers. Furthermore, we proposed constraint memory-enhanced query encoding with (MEQE) to enhance the performance of state-of-the-art neural query encoders on the CEQA task. Our experiments showed that MEQE significantly improved the performance of existing models on the CEQA task. Overall, our work provides a more comprehensive and effective solution to the complex query-answering problem on eventuality knowledge graphs.

## 8 Acknowledgments

The authors of this paper were supported by the NSFC Fund (U20B2053) from the NSFC of China, the RIF (R6020-19 and R6021-20) and the GRF (16211520 and 16205322) from RGC of Hong Kong. We also thank the support from the UGC Research Matching Grants (RMGS20EG01-D, RMGS20CR11, RMGS20CR12, RMGS20EG19, RMGS20EG21, RMGS23CR05, RMGS23EG08).

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

## A    Broader Impact

This paper is the first work discussing how to conduct logical reasoning over knowledge graphs that describe events, states, and actions, known as eventualities. The proposed method, MEQE, is capable of effectively and efficiently answering logical queries over eventuality knowledge graphs.

The experiments were conducted on publicly available knowledge graphs, eliminating any data privacy concerns. However, one possible concern is that our proposed reasoning method is susceptible to adversarial attacks [14, 57, 9] and data poisoning [51] on knowledge graph reasoning systems, which may result in unintended outcomes for users.

## B    Logical Constraints from Discourse Relations

In this paper, we utilize discourse structures based on the early work by Asher [3], where discourse relations are considered as predicates that involve two abstract objects, such as events, states, and propositions [48]. We have adopted the discourse relation definitions from the Penn Discourse Treebank (PDTB) [34], which consist of four general classes: `Temporal`, `Comparison`, `Contingency`, and `Expansion`. Each general class comprises various types, and the logical constraints are derived based on the semantic meaning of these discourse types.

The `Temporal` class is used when there is a temporal relationship between the described situations in the arguments. It includes $\text{Precedence}(A, B)$, $\text{Succession}(A, B)$, and $\text{Synchronous}(A, B)$. In `Temporal` relations, we employ the temporal logic expressions $\succ$, $\prec$, and $=$ to represent the temporal order between two eventualities [22]. $A \prec B$ denotes that A occurs before B, $A = B$ implies that they happen simultaneously, and $A \succ B$ indicates that A occurs after B.

The `Contingency` class is used when one of the described situations in $A$ and $B$ causally influences the other. It encompasses `Reason`, `Result`, and `Condition`. `Reason` describes a cause-and-effect relationship between two eventualities. We use the conditional operator $>$ [21] to represent conditional and causal relations. $\text{Reason}(B, A)$, $\text{Result}(A, B)$, and $\text{Condition}(B, A)$ can all implies $A > B$, indicating that A causes B [21]. Moreover, `Reason` and `Result` also imply they both occur.

The `Comparison` class depicts a discourse relation between $A$ and $B$ to to highlight significant differences between the two situations. Semantically, it indicates that the underlying values of $A$ and $B$ are independent of the connective [35]. Therefore, we simply use $A \wedge B$ to represent both sub-types of $\text{Contrast}(A, B)$ and $\text{Consession}(A, B)$, signifying that both eventualities indeed occur.

`Expansion` class describes those relations that expand the discourse and move its narratives or exposition forward. The $\text{Conjunction}(A, B)$ is used to indicate new situations that provide new information in $B$ that is related to the situation described in $A$. The logical formulation from the conjunction can be expressed as $A \wedge B$. Meanwhile, the $\text{Instantiation}(A, B)$ relation also requires both arguments to hold [35]. Thus it can also be described by the expression $A \wedge B$. $\text{Exception}(A, B)$ indicates that $B$ specifies an exception to the generalization specified by $A$. In other words, $A$ is false because $B$ is true, but if $B$ were false, $A$ would be true. The semantics of an exception is expressed in $\neg A \wedge B \wedge (\neg B \rightarrow A)$. $\text{Restatement}(A, B)$ describes the relationship that the semantics of $B$ restates the semantics of $A$. So the $A$ and $B$ hold true at the same time $A \leftrightarrow B$. $\text{Alternative}(A, B)$ relationship applies when two eventualities describe alternative situations. The semantics of $\text{Alternative}(A, B)$ is $A \vee B$. $\text{ChosenAlternative}(A, B)$ means that two alternatives $A$ and $B$ are given, but the first one $A$ is not chosen. Its semantic meaning is represented as $(A \vee B) \wedge \neg A$.

The `Expansion` class encompasses relations that expand the discourse and advance its narratives or exposition [35]. $\text{Conjunction}(A, B)$ is used to indicate new situations in $B$ that provide related information to the situation described in $A$. The logical formulation from conjunction can be expressed as $A \wedge B$. Similarly, $\text{Instantiation}(A, B)$ also requires both arguments to hold [35] and can be described by the expression $A \wedge B$. $\text{Exception}(A, B)$ indicates that $B$ specifies an exception to the generalization specified by $A$. In other words, $A$ is false because $B$ is true, but if B were false, $A$ would be true. The semantics of an exception can be expressed as $\neg A \wedge B \wedge (\neg B \rightarrow A)$. $\text{Restatement}(A, B)$ describes a relationship where the semantics of $B$ restates the semantics of $A$. Therefore, $A$ and $B$ hold true simultaneously, represented as $A \leftrightarrow B$. $\text{Alternative}(A, B)$ applies

Table 5: The basic information about the ASER-50K used for the experiments, and its standard training, validation, and testing edges separations.

| Dataset | Relation Types | Entities | Training | Validation | Testing | All Edges |
|---------|----------------|----------|----------|------------|---------|-----------|
| ASER50K | 14 | 54,557 | 113,608 | 13,860 | 13,784 | 141,252 |

Table 6: A breakdown of the detailed query types is provided in the training, validation, and testing sets, along with corresponding statistics. Specifically, we report the number of samples, the number of non-contradictory answers, and the number of answers that satisfy the relational constraints but are contradictory due to occurrence or temporal constraints.

| Split | #Anc. | Type | Depths | OccurrenceConstarint # Queries | # Ans. | # Contr. Ans. | Temporal Constraints # Queries | # Ans. | # Contr. Ans. |
|-------|-------|------|--------|----------|--------|---------------|----------|--------|---------------|
| Trn. | 1 | (p,(e)) | 1 | 4,231 | 2.29 | 1.15 | 112 | 3.41 | 1.06 |
| | | (p,(p,(e))) | 2 | 21,010 | 6.03 | 2.09 | 1,876 | 6.16 | 1.36 |
| | 2 | (p,(i,(p,(e)),(p,(e)))) | 2 | 40,728 | 7.59 | 1.63 | 15,941 | 7.44 | 1.20 |
| | | (i,(p,(e)),(p,(e))) | 1 | 3,048 | 1.78 | 1.07 | 84 | 1.60 | 1.00 |
| | | (i,(p,(e)),(p,(p,(e)))) | 2 | 18,088 | 4.93 | 1.38 | 1,940 | 4.10 | 1.10 |
| | | (i,(p,(p,(e))),(p,(p,(e)))) | 2 | 37,661 | 7.50 | 1.87 | 16,009 | 7.43 | 1.19 |
| Val. | 1 | (p,(e)) | 1 | 1,023 | 4.47 | 1.36 | 69 | 4.77 | 1.22 |
| | | (p,(p,(e))) | 2 | 2,317 | 12.82 | 3.33 | 965 | 13.57 | 1.81 |
| | 2 | (p,(i,(p,(e)),(p,(e)))) | 2 | 2,482 | 10.77 | 2.02 | 2,357 | 13.80 | 1.65 |
| | | (i,(p,(e)),(p,(p,(e)))) | 2 | 2,130 | 8.01 | 1.59 | 877 | 8.07 | 1.37 |
| | | (i,(p,(e)),(p,(e))) | 1 | 821 | 2.85 | 1.19 | 71 | 2.08 | 1.21 |
| | | (i,(p,(p,(e))),(p,(p,(e)))) | 2 | 2,391 | 10.13 | 2.32 | 2,338 | 12.50 | 1.50 |
| | 3 | (p,(i,(i,(p,(e)),(p,(e))),(p,(e)))) | 2 | 2,452 | 10.02 | 1.73 | 2,618 | 12.57 | 1.57 |
| | | (i,(p,(e)),(p,(i,(p,(e)),(p,(e))))) | 2 | 2,428 | 9.08 | 1.57 | 2,394 | 10.68 | 1.37 |
| | | (i,(i,(p,(e)),(p,(e))),(p,(e))) | 1 | 1,026 | 2.43 | 1.15 | 281 | 2.20 | 1.23 |
| | | (i,(i,(p,(e)),(p,(p,(e)))),(p,(e))) | 2 | 1,952 | 7.64 | 1.52 | 977 | 8.19 | 1.43 |
| | | (i,(i,(p,(p,(e))),(p,(p,(e)))),(p,(e))) | 2 | 2,327 | 7.89 | 1.59 | 2,368 | 10.86 | 1.39 |
| | | (i,(i,(p,(e)),(p,(p,(e)))),(p,(p,(e)))) | 2 | 2,399 | 9.12 | 1.90 | 2,555 | 11.64 | 1.46 |
| | | (i,(i,(p,(e)),(p,(e))),(p,(p,(e)))) | 2 | 1,862 | 3.10 | 1.30 | 1,068 | 3.67 | 1.44 |
| | | (i,(i,(p,(e)),(p,(p,(e)))),(p,(p,(e)))) | 2 | 2,329 | 7.73 | 1.61 | 2,399 | 10.73 | 1.42 |
| | | (i,(i,(p,(p,(e))),(p,(p,(e)))),(p,(p,(e)))) | 2 | 2,333 | 9.20 | 2.07 | 2,568 | 12.19 | 1.45 |
| Tst. | 1 | (p,(e)) | 1 | 1,091 | 4.83 | 1.38 | 50 | 6.78 | 1.18 |
| | | (p,(p,(e))) | 2 | 2,261 | 14.19 | 3.39 | 954 | 16.50 | 1.85 |
| | 2 | (p,(i,(p,(e)),(p,(e)))) | 2 | 2,425 | 11.77 | 2.20 | 2,434 | 17.13 | 1.95 |
| | | (i,(p,(e)),(p,(e))) | 1 | 899 | 3.29 | 1.23 | 91 | 2.88 | 1.29 |
| | | (i,(p,(e)),(p,(p,(e)))) | 2 | 2,093 | 8.53 | 1.65 | 845 | 10.30 | 1.38 |
| | | (i,(p,(p,(e))),(p,(p,(e)))) | 2 | 2,402 | 10.89 | 2.30 | 2,315 | 15.11 | 1.53 |
| | 3 | (p,(i,(i,(p,(e)),(p,(e))),(p,(e)))) | 2 | 2,386 | 11.26 | 1.81 | 2,648 | 15.95 | 1.77 |
| | | (i,(p,(e)),(p,(i,(p,(e)),(p,(e))))) | 2 | 2,368 | 9.67 | 1.62 | 2,470 | 13.00 | 1.47 |
| | | (i,(i,(p,(e)),(p,(e))),(p,(e))) | 1 | 1,234 | 2.67 | 1.18 | 310 | 2.97 | 1.27 |
| | | (i,(i,(p,(e)),(p,(p,(e)))),(p,(e))) | 2 | 1,928 | 7.76 | 1.55 | 1,049 | 10.24 | 1.45 |
| | | (i,(i,(p,(p,(e))),(p,(p,(e)))),(p,(e))) | 2 | 2,282 | 9.13 | 1.72 | 2,420 | 12.97 | 1.40 |
| | | (i,(p,(i,(p,(e)),(p,(e)))),(p,(p,(e)))) | 2 | 2,346 | 10.09 | 1.93 | 2,607 | 13.89 | 1.64 |
| | | (i,(i,(p,(e)),(p,(e))),(p,(p,(e)))) | 2 | 1,910 | 3.44 | 1.42 | 1,052 | 5.94 | 1.41 |
| | | (i,(i,(p,(e)),(p,(p,(e)))),(p,(p,(e)))) | 2 | 2,297 | 8.33 | 1.63 | 2,423 | 12.68 | 1.39 |
| | | (i,(i,(p,(p,(e))),(p,(p,(e)))),(p,(p,(e)))) | 2 | 2,321 | 10.13 | 2.17 | 2,558 | 14.66 | 1.57 |

when two eventualities describe alternative situations. The semantics of Alternative(A, B) is A ∨ B. ChosenAlternative$(A, B)$ means that two alternatives, $A$ and $B$, are given, but only the first one $A$ is chosen. Its semantic meaning is represented as $(A \lor B) \land \neg B$.

## C Differences Between Commonsense Reasoning and Eventuality Reasoning

Our task is different from other QA or implicit reasoning tasks in several ways. Firstly, it has a broader scope, encompassing various relationships, including non-common sense discourse relations found in Treebank 2.0, which is even challenging for large language models [11]. This resource provides additional relations, which include four general types: temporal (before/after), contingency (because/result), comparison (but/although), and expansions (and/or/except/instead). In contrast, common sense relations mainly focus on the first two types of relations: contingency and temporal. The occurrence constraints discussed in this paper primarily exist in the expansion type, which does not appear in common sense KG but exists in the event KG. This makes our task more complex, and it cannot be effectively addressed using common sense question-answering methods [44].

Moreover, our main focus is on complex query answering, where queries center around intricate relationships between eventualities. Unlike existing common sense knowledge graphs (CSKGs), which typically handle relations involving **two** events in a triple, our task involves **multiple** events within a single query-answer pair. This presents a challenge in formulating our task as either a knowledge graph completion (KGC) or question-answering (QA) task, as such formulations would require discarding most query constraints, reducing complexity, and simplifying it into a basic query answering task. While commonsense knowledge may play a role in answering our queries, it is not as prevalent as in other tasks [45]. Additionally, our task does not heavily rely on the semantic information of the query itself; instead, it relies on learning graph structures to perform query answering and reasoning. We utilize the inherent structure of the graph rather than relying solely on natural language processing. Finally, there are several complex query-answering tasks that share similar settings with the one in our paper, such as the EFO-1 benchmarks [46].

# D   Knowledge Graph Details

The eventuality knowledge graph, ASER-50K, is derived from a sub-sample of ASER2.1[4]. ASER2.1 includes the `Co-Occurrence` relations, which indicate that two eventualities co-occur in two consecutive sentences in the original text. However, in this paper, we exclude the co-occurrence relation to focus on discourse relations. To remove noise from ASER 2.1, we eliminate edges with low frequencies and retain only those with a frequency higher than two. The ASER graph is constructed using an extractive method from natural language text, which may result in the inclusion of eventualities with high frequency but vague semantics, such as *PersonX know* and *PersonX think*. To address this issue, we remove the most frequent one hundred vertices and retain the remaining densest vertices. The resulting ASER-50K dataset comprises 54,557 eventualities and 141,252 edges. Subsequently, we randomly partition the edges into training, evaluation, and testing sets in an 8:1:1 ratio. The numbers of edges in each set are presented in Table 5.

The query types in our framework reflect the structure of the computational graph and are represented using a Lisp-like format [46, 7]. For instance, the query `(i,(p,(e)),(p,(e)))` represents a query with two anchor eventualities, each having a relational projection, and the answer eventualities are the intersection of these two projection results. Additionally, this query type is also referred to as `2i` in related work [37, 36]. However, our naming approach is more flexible and can be extended to accommodate more complex query structures. We sample our queries based on the query types, limiting them to a maximum of three anchors and a maximum depth of two. Specifically, in the training set, we only sample queries with a maximum of two anchors. Further details regarding the query types in the training, validation, and testing sets can be found in Table 6.

# E   Query Sampling Algorithm

The query sampling algorithm employed in this study is based on the work by Ren et al. [37]. We replicate the sampling algorithm and provide the pseudo-code for the sampling process in Algorithm 1. Our focus in this paper is on conjunctive logical queries derived from eventuality knowledge graphs. As a result, the query sampling process involves only the operations of *relational projections* and *intersections*. Given a knowledge graph $G$ and a query type $T$, we initiate the query generation process by starting with a random node $v$. The goal is to recursively construct a query that has $v$ as its answer, following the structure specified by $T$. During each recursive step, we examine the last operation in the query. If the operation is a *projection*, we randomly select one of its predecessors $u$ that holds the corresponding relation to $v$, which will serve as the answer to the sub-query. The recursion is then applied to node $u$ and the sub-query type of $T$. Similarly, for *intersection*, we recursively apply the process to their respective sub-queries on the same node $v$. The recursion continues until the current node contains an anchor entity, at which point the process terminates. This recursive approach allows us to systematically construct queries that satisfy the given query type $T$ and have $v$ as the desired answer.

---

[4]https://hkust-knowcomp.github.io/ASER/html/index.html

**Algorithm 1** The algorithm used for sampling a complex query from a knowledge graph starting from a random vertex $v$ from the knowledge graph $G$ with query structure $T$.

---
**Require:** $G$ is a knowledge graph.
  **function** SAMPLEQUERY($T, v$)
      $T$ is an arbitrary node of the computation graph.
      $v$ is an arbitrary knowledge graph vertex
      **if** $T.operation = p$ **then**
         $u \leftarrow$ SAMPLE($\{u | (u, v)\text{is an edge in } G\}$)
         $RelType \leftarrow$ type of $(u, v)$ in $G$
         $ProjectionType \leftarrow p$
         $SubQuery \leftarrow$ SAMPLEQUERY($T.child, u$)
         **return** $(ProjectionType, RelType, SubQuery)$
      **else if** $T.operation = i$ **then**
         $IntersectionResult \leftarrow (i)$
         **for** $child \in T.Children$ **do**
            $SubQuery \leftarrow$ SAMPLEQUERY($T.child, v$)
            $IntersectionResult$.PUSHBACK($child, v$)
         **end for**
         **return** $IntersectionResult$
      **else if** $T.operation = e$ **then**
         **return** $(e, T.value)$
      **end if**
  **end function**

---

Table 7: Ablation Studies on Constraints and Feed-Forward Network in MEQE.

| Models | Occurrence Constraints | | | Temporal Constraints | | | Average | | |
|---|---|---|---|---|---|---|---|---|---|
| | Hit@1 | Hit@3 | MRR | Hit@1 | Hit@3 | MRR | Hit@1 | Hit@3 | MRR |
| GQE | 8.92 | 14.21 | 13.09 | 9.09 | 14.03 | 12.94 | 9.12 | 14.12 | 13.02 |
| + MEQE | **10.20** | **15.54** | **14.31** | **10.70** | **15.67** | **14.50** | **10.45** | **15.60** | **14.41** |
| + MEQE - Constraints | 8.29 | 12.87 | 11.62 | 8.80 | 13.02 | 12.17 | 8.54 | 12.95 | 11.90 |
| + MEQE - FFN | 0.67 | 1.17 | 1.13 | 0.74 | 1.23 | 1.12 | 0.70 | 1.19 | 1.08 |
| Q2P | 14.14 | 19.97 | 18.84 | 14.48 | 19.69 | 18.68 | 14.31 | 19.83 | 18.76 |
| + MEQE | **15.15** | **20.67** | **19.38** | **16.06** | **20.82** | **19.74** | **15.61** | **20.74** | **19.56** |
| + MEQE - Constraints | 14.16 | 20.00 | 18.86 | 14.72 | 19.92 | 18.79 | 14.44 | 19.96 | 18.82 |
| + MEQE - FFN | 12.77 | 16.63 | 15.89 | 12.74 | 16.83 | 14.75 | 12.76 | 16.73 | 15.32 |
| Nerual MLP | 13.03 | 19.21 | 17.75 | 13.45 | 19.06 | 17.68 | 13.24 | 19.14 | 17.71 |
| + MEQE | **15.26** | **20.69** | **19.32** | **15.91** | **20.63** | **19.47** | **15.58** | **20.66** | **19.40** |
| + MEQE - Constraints | 13.33 | 19.15 | 17.94 | 13.49 | 19.18 | 14.48 | 13.41 | 19.16 | 18.08 |
| + MEQE - FFN | 10.35 | 14.67 | 13.71 | 10.94 | 14.67 | 12.74 | 10.64 | 14.67 | 14.53 |
| FuzzQE | 11.68 | 18.64 | 17.07 | 11.68 | 17.97 | 16.53 | 11.68 | 18.31 | 16.80 |
| + MEQE | **14.76** | **21.12** | **19.45** | **15.31** | **21.01** | **19.49** | **15.03** | **21.06** | **19.47** |
| + MEQE - Constraints | 12.69 | 19.92 | 17.68 | 13.53 | 18.25 | 17.91 | 13.11 | 19.08 | 17.80 |
| + MEQE - FFN | 9.81 | 15.26 | 14.46 | 10.17 | 15.37 | 14.87 | 9.99 | 15.31 | 14.66 |

## F   Further Ablation Study on the memory module and FFN layer

To demonstrate the effectiveness of the relevance score and the feed-forward module, we conducted an ablation study on our proposed MEQE method, and the results are presented below.

When we removed the feed-forward network, as shown in the rows of "MEQE - FFN" and directly added the relations and tails embedding to the query embedding, the performance was negatively impacted. This is because the query embedding is more likely to have a higher similarity to the answers that should be excluded. This effect was more significant in the GQE model, as the GQE model uses the simplest operation as the relations projection.

We also conducted another ablation study by replacing the constraints with random triples so that there are no contradictory answers in the rows of "MEQE-Constraints". We observed that the performance of the model is comparable to the baseline model. This indicates that the performance improvement is gained from the constraints instead of the structural changes of the query encoder.

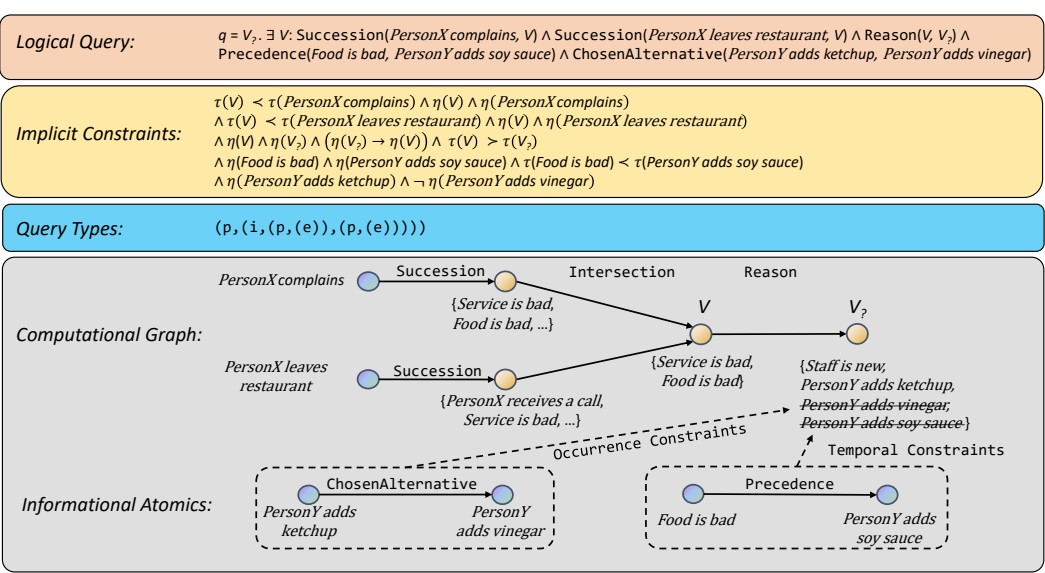

Figure 5: The example provided showcases a complex eventuality query along with its implicit constraints, query type, computational graph and atomics visualization.

These experiments prove two things. First, the relevance score is effective in finding the corresponding constraints. Second, the feed-forward layer is useful and necessary to adjust the direction of the memory contents to incorporate into the query embedding.

## G    Detailed Example of Complex Eventuality Query

Figure 5 provides a detailed example of a complex eventuality query. This query corresponds to the query type (p,(i,(p,(e)),(p,(e)))), and its corresponding computational graph is depicted. The implicit constraints of the atomics in the logical query are derived according to the discourse relations. When the computational graph is executed on the eventuality knowledge graph, without considering the logical constraints, there would be four potential answers: *Staff is new, PersonY adds ketchup, PersonY adds vinegar*, and *PersonY adds soy sauce*.

However, the answer *PersonY adds vinegar* is contradictory due to occurrence constraints, as one of the informational atomics indicates that *PersonY adds vinegar* did not occur. Furthermore, the answer *PersonY adds soy sauce* is contradictory due to temporal constraints, as it occurs after *Food is bad*, indicating that *PersonY adds soy sauce* cannot be the reason for *Food is bad*.

