# OpenReview forum: "Complex Query Answering on Eventuality Knowledge Graph with Implicit Logical Constraints"
_NeurIPS.cc/2023/Conference — NeurIPS 2023 poster_

### Official Review · Reviewer_YBho · 2023-06-25

**Soundness:** 3 good
**Presentation:** 2 fair
**Contribution:** 3 good
**Rating:** 5
**Confidence:** 4

**Summary:**

The paper extends the traditional complex query-answering task into an eventuality-centric complex query-answering task to understand the reasoning at the eventuality level. Specifically, the paper divides the discourse rations into two types of implicit constraints: occurrence constraints and temporal constraints. The occurrence constraints determine whether certain eventuality happens or not. The temporal constraints provide the order of the eventualities' occurrence. In addition, the paper proposes a new memory-enhanced query encoding to reason based on eventuality-centric knowledge graphs. The model first introduces a computational graph that encodes queries, including operations such as relational projection and intersections. The paper proposes a memory-enhanced encoding component that utilizes a memory module to encode constraint information. The operation output is used as a query to access each head eventuality based on the relevance score. The paper then uses an attentional aggregation to sum over the constraint relation and tails. The final model is optimized based on similarity scores with cross-entropy loss.

**Strengths:**

1. The paper introduces a new eventuality-centric complex query-answering task to better model the reasoning at the eventuality level. The paper proposes a new way to provide discourse relations with two implicit logic constraints. The idea of occurrence constraints and temporal constraints is interesting.
2. The paper introduces a new memory-enhanced query encoding to update the query representation with relevance-based constraint representations.
3. The paper tests the new framework with a new dataset sampled from ASER. The model shows strong performance over multiple different baselines. The paper also provides code and dataset construction details in the Appendix. The paper also includes a case study in the Appendix.

**Weaknesses:**

Some parts of the paper are not very clear:
1. In section 3.1, what function is used for relation projection and intersection operation? The paper said that the intersection is a permutation-invariant neural network. However, it needs to be clarified in detail to readers.
2. In section 3.2, what relevance score is used for Equation 5? Suppose the paper used a semantic relevance score such as cosine similarity. In that case, the motivation for this part is unclear because the constraints with higher similarity might not be the ones with closed relevance. Moreover, Figure 4 needs to be clarified. The paper needs to briefly explain the right part (constraint memory in Figure 4.) I suggest going through the walkthrough example in section 3.2. The code does not include a ReadMe file.
3. In section 4.1, the paper only focuses on the answer with constraints. Has the paper also tested the new model for answers without any contradictions?
4. The abbreviation in Table 4 needs to be clarified. It seems that p is the projection, i is the intersection. However, those abbreviations are not clarified in the caption. Readers also find it hard to figure out what e represents. The analysis of Table 4 is superficial. The paper needs to add more qualitative analysis with more concrete examples. The paper needs to conduct an ablation study to show the contribution of each component.

**Questions:**

1. In section 3.1, what function is used for relation projection and intersection operation?
2. In section 3.2, what relevance score is used for Equation 5?
3. Has the paper also tested the new model for answers without any contradictions?

**Limitations:**

The paper provides a limitation section and broader impact in the Appendix.

---

> ### Author Rebuttal · Authors · 2023-08-10
>
> Re W1 & Q1:
>
> The relation projections and intersection operations are adopted from backbone models. For different backbone bone models, we have different parametrization of Intersection/Union.
>
> GQE [1] model uses a feed-forward layer, followed by an average pooling, and then followed by a matrix multiplication for intersection. The relational projection is modeled by using a matrix multiplication.
> Query2Particles [2] use self-attentions for intersection modeling, and the gated transition function for relational projection.
> FuzzQE [3] uses element-wise fuzzy logic operations on embedding space to do intersections and unions. Meanwhile it use a feed forward layer, a layer normalization, and a sigmoid function for relational projection.
> Neural-MLP [4] uses MLP-Mixer as an intersection module and the relation projection module.
>
>
>
> Re W2 & Q2:
>
> All existing query encoders use the similarity between the query embedding and answers embedding to retrieve answers. In our case, this similarity score is an inner product. Because of this, we can use the same similarity measure to compute the relevance between the query embedding and the head eventuality embedding as a relevance score to the constraint. If the constraint is relevant to the query embedding, then relation and tail information of this constraint are added into the query representation.
>
> However, if a constraint is relevant, we want to exclude the information of its relation and tail from the query embedding, we are motivated to add a feed-forward layer to adjust the direction of the memory value, which contains the relation and tail embedding.
>
> For example, in the example of Figure 4, the “Food is bad” in the constraint memory has a high relevance score to the “the events happens before X complains and leaves the restaurant”.
>
> Then, when the constraint of “Food is bad before PersonY adds soy sauce” is given, we add the information of relation type “Precedence” and tail node “PersonY adds soy sauce” from memory value to the query embedding.  We are motivated to do this because we want to exclude the answer of  “PersonY adds soy sauce” if the following relation projection is “Reason”, “Condition”, or “Succession”, because we want to avoid temporal contradiction.
> We further prove the correctness of the intuition behind this idea by the ablation study conducted in Re W4.
>
>
> Re W3 & Q3:
>
> We do not fully this question, because “the answers with constraints” has the same meaning as “the answers without contradictions”. We filtered out the contradictory answers according to the theorem prover, and only kept the answers that satisfy the logical constraints in the dataset construction process.
>
>
> | Models |  | Occurence Constaints |  |  | Temporal Constraints |  |  | Average |  |  |  |
> |---|:---:|:---:|:---:|:---:|:---:|:---:|:---:|:---:|:---:|---|---|
> |  | Hit@1 | Hit@3 | MRR | Hit@1 | Hit@3 | MRR | Hit@1 | Hit@3 | MRR |  |  |
> | GQE | 8.92 | 14.21 | 13.09 | 9.09 | 14.03 | 12.94 | 9.12 | 14.12 | 13.02 |  |  |
> | + CMQE | 10.20 | 15.54 | 14.31 | 10.70 | 15.67 | 14.50 | 10.45 | 15.60 | 14.41 |  |  |
> | + CMQE - Constraints | 8.29 | 12.87 | 11.62 | 8.80 | 13.02 | 12.17 | 8.54 | 12.95 | 11.90 |  |  |
> | + CMQE - FFN | 0.67 | 1.17 | 1.13 | 0.74 | 1.23 | 1.12 | 0.70 | 1.19 | 1.08 |  |  |
> | Q2P | 14.14 | 19.97 | 18.84 | 14.48 | 19.69 | 18.68 | 14.31 | 19.83 | 18.76 |  |  |
> | + CMQE | 15.15 | 20.67 | 19.38 | 16.06 | 20.82 | 19.74 | 15.61 | 20.74 | 19.56 |  |  |
> | + CMQE - Constraints | 14.16 | 20.00 | 18.86 | 14.72 | 19.92 | 18.79 | 14.44 | 19.96 | 18.82 |  |  |
> | + CMQE - FFN | 12.77 | 16.63 | 15.89 | 12.74 | 16.83 | 14.75 | 12.76 | 16.73 | 15.32 |  |  |
> | Nerual MLP | 13.03 | 19.21 | 17.75 | 13.45 | 19.06 | 17.68 | 13.24 | 19.14 | 17.71 |  |  |
> | + CMQE | 15.26 | 20.69 | 19.32 | 15.91 | 20.63 | 19.47 | 15.58 | 20.66 | 19.40 |  |  |
> | + CMQE - Constraints | 13.33 | 19.15 | 17.94 | 13.49 | 19.18 | 14.48 | 13.41 | 19.16 | 18.08 |  |  |
> |  + CMQE - FFN | 10.35 | 14.67 | 13.71 | 10.94 | 14.67 | 12.74 | 10.64 | 14.67 | 14.53 |  |  |
> | FuzzQE | 11.68 | 18.64 | 17.07 | 11.68 | 17.97 | 16.53 | 11.68 | 18.31 | 16.80 |  |  |
> | + CMQE | 14.76 | 21.12 | 19.45 | 15.31 | 21.01 | 19.49 | 15.03 | 21.06 | 19.47 |  |  |
> | + CMQE - Constraints | 12.69 | 19.92 | 17.68 | 13.53 | 18.25 | 17.91 | 13.11 | 19.08 | 17.80 |  |  |
> | + CMQE - FFN | 9.81 | 15.26 | 14.46 | 10.17 | 15.37 | 14.87 | 9.99 | 15.31 | 14.66 |  |  |
>
>
>
> Re W4:
>
> Yes, for the query types notations, the “p” is for projection, the “i” is for intersection, and “e” is for eventuality. We will include the detailed introduction of query types and abbreviations in the paper.
>
> We conducted an ablation study on our proposed MEQE method to prove the effectiveness of the relevance score and the feed-forward To prove the effectiveness of the relevance score and the feed-forward module mentioned in Re W2, we conducted an ablation study on our proposed MEQE method, and here are the results: When we removed the feed-forward network and directly added the relations and tails embedding to the query embedding, the performance was negatively affected. This is because the query embedding is more likely to have higher similarity to the answers that should be excluded. This effect was more significant in the GQE model, as the GQE model uses the simplest element-wise addition as the relations projection.
>
> We conduct another ablation of replacing the constraints to random triples so that there are no contradictory answers. Then we observed that the performance of the baseline models is comparable with the MEQE model. This indicates that the performance improvement is gained from the constraints instead of the structural changes of the query encoder.
>
> This experiment proves two things. First, the relevance score is effective in finding the corresponding constraints. Second, the feed-forward layer is useful and necessary to adjust the direction of the memory contents to incorporate into the query embedding.

---

> > ### Comment · Reviewer_YBho · 2023-08-13
> > **Thanks for the rebuttal**
> >
> > The authors have answered most of my questions. I have raised my score from 4 to 5.
> >
> > I want to clarify the W3 and Q3. So basically, I wonder about the performance of the dataset without any constraints. Will the model cause a performance drop in those instances?

---

> > > ### Author Response · Authors · 2023-08-19
> > >
> > > Thank you for your reply and clarification. To address the issue you raised, we have sampled another round of evaluation data with informational atomics (i.e., contents in the memory module) that do not have constraints to the answers. We have ensured that there are no contradictory answers in any of these instances.
> > >
> > > The performances are as follows:
> > >
> > > | Models |  | Occurrence Constraints |  |  | Temporal Constraints |  |  | Average |  |
> > > |---|:---:|:---:|:---:|:---:|:---:|:---:|:---:|:---:|:---:|
> > > |  | Hit@1 | Hit@3 | MRR | Hit@1 | Hit@3 | MRR | Hit@1 | Hit@3 | MRR |
> > > | GQE | 10.06 | 15.87 | 14.51 | 9.56 | 14.68 | 13.68 | 9.81 | 15.27 | 14.10 |
> > > |  + CMQE | 10.98 | 16.87 | 15.37 | 11.03 | 15.34 | 14.41 | 11.00 | 16.10 | 14.89 |
> > > | Q2P | 11.94 | 17.61 | 16.11 | 11.36 | 16.69 | 14.89 | 11.65 | 17.15 | 15.50 |
> > > |  + CMQE | 13.13 | 17.78 | 16.53 | 12.72 | 16.78 | 15.75 | 12.93 | 17.28 | 16.14 |
> > > | Neural MLP | 16.58 | 22.00 | 20.78 | 16.24 | 21.15 | 20.09 | 15.23 | 21.57 | 20.43 |
> > > |  + CMQE | 16.88 | 21.60 | 20.38 | 16.85 | 20.93 | 19.91 | 15.52 | 21.26 | 20.14 |
> > > | FuzzQE | 15.71 | 22.17 | 20.44 | 15.15 | 21.61 | 19.42 | 15.43 | 21.89 | 19.93 |
> > > |  + CMQE | 16.50 | 23.01 | 21.10 | 15.46 | 20.70 | 19.75 | 15.98 | 21.86 | 20.43 |
> > >
> > > Generally, the performance of these models is comparable. However, MEQE performs slightly better when used together with GQE, FuzzQE, and Q2P, while it is comparable to Neural MLP.
> > >
> > > Although the memory contents do not have constraints on the query answers, the subtle performance improvement can be explained from two perspectives. First, the informational atomics are sampled from the edges related to the queries, providing additional information about the entities in the query, even though they do not have a direct impact on the answers. Second, MEQE has more parameters than the baseline models. However, as we explained in our previous reply, the structural changes are not the main reason for the performance improvement, as shown in our ablation study.
> > >
> > > We hope that this explanation clarifies the performance comparison between the backbone and MEQE models.
> > >
> > > Thank you again for your valuable feedback.

---

> > > > ### Comment · Reviewer_YBho · 2023-08-19
> > > > **Response to authors' rebuttal**
> > > >
> > > > Thank you so much for your clarification!

---

### Official Review · Reviewer_Ne3e · 2023-07-07

**Soundness:** 3 good
**Presentation:** 4 excellent
**Contribution:** 3 good
**Rating:** 6
**Confidence:** 3

**Summary:**

The paper proposed a reasoning task "Complex Eventuality Query Answering (CEQA)". CEQA is performed over EVKG and is different from traditional CQA over entity-centric KG. Authors of the paper clearly explain the new task, and further discussed a memory-augmented method to improve models' performance on CEQA.

**Strengths:**

The paper made two contributions. First, it proposed the CEQA task and discussed its difference from the traditional CQA task. Then, the paper proposed a memory augmentation method for query encoding.

The authors clearly discussed the CEQA task and emphasized its importance in logical reasoning.

**Weaknesses:**

I have a few quick comments of the paper.

Maybe it worth briefly discussing System 1 and 2 for the completeness of the paper. Readers may be unfamiliar with the terms.

Please consider discussing your dataset created from ASER earlier in the paper and maybe go through a few examples. This task is new, to my knowledge, to many readers including myself. Maybe also discuss ASER a bit more. How is ASER constructed?

I have another concern about the quality of ASER, the backbone database of the proposed task. Some statements and/or reasoning can be ambiguous or debatable. For example, someone can say, "PersonX adds soy sauce" is the cause of "Food is bad" due to some other implicit information in addition to the temporal one. How did ASER (or other similar database resolve this problem)?

How is ASER related and different from Commonsense QA? Both require reasoning with implicit information. Are there other relevant tasks in addition to the ones you described in the paper. Please include them in the Related Work section.

Are you able to experiment with a few more datasets? I understand this is a new task with limited resources. However, experiment results in this paper is limited to fully prove the effectiveness of your proposed method.


**Questions:**

Please see above.

**Limitations:**

The paper proposed a novel reasoning task potentially useful for building more powerful and general reasoning systems. The paper has made substantial contribution in proposing new tasks, but is limited in modeling and experiments.

---

> ### Author Rebuttal · Authors · 2023-08-09
>
> Thanks for your suggestions and efforts for reviewing this paper.
>
> W1 System 1 and System 2:
>
> In short, the theory of System 1 and System 2 reasoning, proposed by Daniel Kahneman, suggests that human thinking can be divided into two systems. System 1 operates automatically and intuitively, making quick judgments and performing routine tasks effortlessly. System 2, on the other hand, engages in deliberate and analytical thinking, requiring conscious effort. In the context of logical query answering, System 2 reasoning is engaged when we need to carefully analyze the question, consider relevant information, and apply logical rules to arrive at a correct answer. We will incorporate brief discussions in our introduction later.
>
>
>
> W2 curated benchmark and ASER:
>
> We will add a subsection presenting more detailed statistics, knowledge format, and examples of ASER and our constructed benchmark. For your information, ASER is constructed via two main steps: extraction and conceptualization. The extraction step involves extracting eventualities (events or situations) from a collection of large corpora. Then, in the conceptualization step, the authors use a graph-based approach to represent the extracted eventualities and their relations as a knowledge graph. This involves identifying and linking semantically related eventualities and relations to form a large-scale eventuality knowledge graph.
>
> W3 :
>
> The question posed is a good one, and to answer it, we need to understand how the edges of ASER are constructed from text using information extraction methods. When a statement like "PersonX adds soy sauce" is identified as the cause of "Food is bad," and both phrases appear in the text corpus above a certain frequency, they are recorded in the ASER graph. Similarly, phrases like "PersonX sleeps" before and after "PersonX takes a shower" can both be plausible high-frequency edges that are stored in the KG, despite being contradictory in a specific situation.
> It's important to note that, in a random subgraph of ASER or a similar database, not all edges can hold simultaneously in a specific situation. This is a characteristic of a database describing events and activities, and not a shortcoming of the database itself. The KG is designed to provide all possible answer candidates, while logical verification is left to the reasoning process. This is the main motivation behind the approach taken in this paper.
>
>
> W4 Difference with commonsense QA:
>
> As we discussed with Reviewer DAya, our task different with other QA or implicit reasoning tasks. Our task is distinct from traditional question answering tasks due to its wide scope, encompassing various relationships, including non-common sense relations found in Treebank 2.0. This resource provides additional relations, making our task more complex. In particular, we encounter queries that involve relationships unique to eventuality levels, such as co-occurrence, conjunction, and contradiction. These intricate connections cannot be effectively addressed using common sense question answering methods. Our main focus is on complex query answering, where queries center around intricate relationships between eventualities. Unlike existing common sense knowledge graphs (CSKGs), which typically handle relations involving two events in a triple, our task involves multiple events within a single query-answer pair. This presents a challenge in formulating our task as either a knowledge graph completion (KGC) or question-answering (QA) task, as such formulations would require discarding most query constraints, reducing complexity, and simplifying it into a basic query answering task. While commonsense knowledge may play a role in answering our queries, it is not as prevalent as in other tasks. Additionally, our task does not heavily rely on the semantic information of the query itself; instead, it relies on learning graph structures to perform query answering and reasoning. We utilize the inherent structure of the graph rather than relying solely on natural language processing.
> As for relevant tasks, there are several complex query answering tasks that share similar settings with the one in our paper, like the EFO1 benchmarks [2].
> We will make sure these differences are more clearly discussed in our paper.
>
> W5 More dataset:
>
> Thank you for your suggestion. Incorporating additional datasets would undoubtedly strengthen our paper. However, at this stage, we have been unable to identify other viable datasets. The main challenge lies in finding knowledge bases or graphs that encompass both (i) eventualities, including states, actions, and events, as nodes, and (ii) comprehensive relationships between these eventualities, as edges. We will remain vigilant and explore any new resources that become available in the future.
>
>
> Reference:
>
> [1] Daniel, K. (2017). Thinking, fast and slow.
>
> [2] Wang, Z., Yin, H., & Song, Y. (2022). Benchmarking the Combinatorial Generalizability of Complex Query Answering on Knowledge Graphs. Proceedings of the Neural Information Processing Systems Track on Datasets and Benchmarks 1 (NeurIPS Datasets and Benchmarks 2021).

---

### Official Review · Reviewer_dKK5 · 2023-07-07

**Soundness:** 4 excellent
**Presentation:** 3 good
**Contribution:** 3 good
**Rating:** 6
**Confidence:** 3

**Summary:**

This work aims to conduct complex logical query task over eventuality-centric KG (EVKG) and propose the Complex Eventuality Query Answering (CEQA) setting that considers the implicit constraint of the temporal order and occurrence of eventualities. The authors also propose a memory-enhanced query encoding method and achieve state-of-the-art performance on the CEQA task.

**Strengths:**

1. very interesting and important research problem
2. the proposed MEQE module enhances the state-of-the-art query encoder on the CEQA task
3. the paper is overall well-organized and well-written

**Weaknesses:**

1. the novelty of memory-enhanced module is limited since the memory mechanism has been proposed in many prior works such as [1]
2. Recently, path-based KG reasoning methods such as QE-GNN [2] have achieved much research progress and show stronger reasoning ability than embedding-based methods. However, the proposed MEQE seems cannot incorporate the path-based method that didn't learn the representation for each node and relation in KGs.

[1] Rossi, E., Chamberlain, B., Frasca, F., Eynard, D., Monti, F., & Bronstein, M. (2020). Temporal graph networks for deep learning on dynamic graphs. arXiv preprint arXiv:2006.10637.
[2] Zhu, Z., Galkin, M., Zhang, Z., & Tang, J. (2022, June). Neural-symbolic models for logical queries on knowledge graphs. In International Conference on Machine Learning (pp. 27454-27478). PMLR.

**Questions:**

please refer to the weakness

---

> ### Author Rebuttal · Authors · 2023-08-09
>
> Re W1:
>
> Thank you for pointing out this reference, and we will cite the corresponding papers. Meanwhile, we would like to argue that, we are the first work to use memory modules in the problem of query encoding, and we are the first work to propose using memory modules to encode the logical constraints during the query encoding process. This is a novel idea worth noticing in the community of knowledge graph reasoning.
>
> Re W2:
>
> Yes, we admit that the current design of our approach cannot be directly used together with GNN-QE. In the current research context, it is the only method that uses GNN over the underlying knowledge graph.
>
> However, GNN-QE obtained superior performance at a high computational cost. Here are the reasons:
>
> 1. The projection operation of GNN-QE relies on GNN operations, and with time complexity $O(|V|d^2 + |E|d)$, where $|V|$ is the number of vertices, $|E|$ is the number of edges, and d is GNN hidden size. The inference time grows linearly with the size of KG. However, the projection operations of other QE methods are of $O(d^2)$, and they are independent of KG size.
>
> 2. The query representation size of GNN-QE is $|V|$ instead of $d$ as previous methods. In previous work, their query embedding size is around 15,000 (the number of vertices),  while other QE methods have the query embedding size at around 300-400.
>
> Moreover, for the previous query encoding methods, they can be scaled-up to the graph with 86,000,000 nodes [1]. Yet, we are not confident that the QE-GNN can also scale up to large graphs.  Meanwhile, we also evaluate the performance of FuzzQE, another fuzzy logical based method over the fixed query embedding size.
>
> [1] Ren, H., Dai, H., Dai, B., Chen, X., Zhou, D., Leskovec, J., & Schuurmans, D. (2022, August). Smore: Knowledge graph completion and multi-hop reasoning in massive knowledge graphs. In Proceedings of the 28th ACM SIGKDD Conference on Knowledge Discovery and Data Mining (pp. 1472-1482).

---

### Official Review · Reviewer_ktzN · 2023-07-07

**Soundness:** 3 good
**Presentation:** 3 good
**Contribution:** 3 good
**Rating:** 5
**Confidence:** 3

**Summary:**

This paper proposes an approach to address the challenge of complex query answering on eventuality knowledge graphs by integrating implicit logical constraints. The authors introduce the task of complex eventuality query answering (CEQA), which requires considering the occurrence and temporal order of eventualities. Methodologically, the paper encodes the edges of the knowledge graph containing these constraints as key-value pairs, which are then integrated into the attention mechanism. The authors extracted eligible data from the ASER dataset and conducted experiments combining their proposed method with various query encoding models, demonstrating improved performance.

**Strengths:**

- The paper's motivation is solid, aiming to incorporate logical information from eventualities into complex query answers.
- The method serves as an effective additional information exploitation approach, adaptable to various query encoding models.
- The constructed CEQA dataset can provide insights for related tasks.

**Weaknesses:**

- This paper dedicates excessive description to the introduction. The use of multiple representations of first-order logic may be misleading, as the actual encoding is relational.
- The work assumes the existence of a knowledge graph and constructs datasets using theorem provers, limiting its generalizability to other tasks.
- The paper introduces two types of constraints and presents contradictory answers in the data, but lacks analysis of these categories in the methods and experiments.  While Table 2 provides statistics on constraints and contradictory answers, conducting further experiments specifically targeting these categories would provide valuable insights into the functioning of the model.

**Questions:**

- In Figure 3, please provide a brief explanation of the confliction of the "Precedence" relation in the two subfigures.
- Could you provide a concise overview of the structure of the permutation-invariant neural network for the intersection operation in Section 3.1?
- Consider reducing the number of method-independent first-order logic examples and representations in order to allocate more space for comprehensive experimental and methodological analyses of the defined categories in the paper.


**Limitations:**

I have no concerns about the limitations.

---

> ### Author Rebuttal · Authors · 2023-08-10
>
> Thank you for your review! I would like to address your concerns one by one.
>
> Re: W1
>
> We clarify that the proposed problem is formally defined in logical form. Meanwhile, relational encoding is part of the query encoding method. It is possible that there will be other methods for this task that do not use relations encoding.
>
> Moreover, the definition in logical form is necessary, because we are using the logical expression as input to the theorem prover to verify the correctness of the answers. This is an important part of our problem definition.
>
>
> Re: W2
>
> In this paper, our proposed method is a logical query answering method on KG, so we do need a knowledge graph as it is part of our problem definition.
>
> However, we argue that it is a novel and generalizable idea to use theorem prover for verifying the consistency of the relations among multiple events. This is because events can also be obtained from other sources, for example text, instead of sampling from a KG.
>
> For a specific example, we can use this idea to verify the logical consistency of the stories generated from language models. In doing so, we can first extract events from the stories, and use discourse parsers to create edges among the events. Then we can apply the theorem prover on this graph of events to verify whether there are contradictions on occurrence and the order of occurrence of events.
>
>
> Re:W3
>
> We would like to clarify that we do indeed include the corresponding results for the two types of constraints and contradictory answers in Table 3. This table has nine columns, with the three columns on the left side showing the results with occurrence constraints, the three columns in the middle showing the results on temporal constraints. The last three columns show the averaged results.
>
>
>
> Re: Q1
>
> In table 3, V is something that happens before a person complains and leaves the restaurant, according to the KG, the V could be either “Service is bad” or “Food is bad”. If V? is the reason of V, then according to the graph, V? could be either “Staff is new”, “PersonY adds ketchup”, “PersonY adds soy sauce”, and “PersonY adds vinegar”.
>
> However, in the query we also know that  “PersonY adds vinegar” does not happen, and “PersonY adds soy sauce” happens after the “Food is bad”, thus cannot be the reason for “Food is bad”. The conflict here is causality implies presidency, and an specific event cannot happen both before and after another event.
>
>
> Re: Q2
>
> The relation projections and intersection operations are adopted from backbone models. For different backbone bone models,  we have different parametrization of Intersection/Union. But basically, they are all deep-set functions [1].
>
> * GQE [2] model uses a feed-forward layer, followed by an average pooling, and then followed by a matrix multiplication.
> * Query2Particles [3] use self-attentions for intersection modeling:
> * FuzzQE [4] uses element-wise fuzzy logic operations on embedding space to do intersections and unions.
> * Neural-MLP [5] uses MLP-Mixer as an intersection module.
>
> All the baseline models have their unique designs, but they share one thing in common, their intersection operations are all invariant to the permutation. They are all special types of deep set functions [1]. We will include the formula of them to the appendix of this paper.
>
>
> Re: Q3
>
> Thank you for your advice, we will move parts of the logical definition to the appendix, so that we can include the newly added experiment results to the paper.
>
>
> Reference:
>
> [1] Zaheer, M., Kottur, S., Ravanbakhsh, S., Poczos, B., Salakhutdinov, R. R., & Smola, A. J. (2017). Deep sets. Advances in neural information processing systems, 30.
>
> [2] Hamilton, Will, et al. "Embedding logical queries on knowledge graphs." Advances in neural information processing systems 31 (2018).
>
> [3] Jiaxin Bai, Zihao Wang, Hongming Zhang, and Yangqiu Song. 2022. Query2Particles: Knowledge Graph Reasoning with Particle Embeddings. In Findings of the Association for Computational Linguistics: NAACL 2022. Association for Computational Linguistics, Seattle, United States, 2703–2714.
>
> [4] Chen, Xuelu, Ziniu Hu, and Yizhou Sun. "Fuzzy logic based logical query answering on knowledge graphs." Proceedings of the AAAI Conference on Artificial Intelligence. Vol. 36. No. 4. 2022.
>
> [5] Alfonso Amayuelas, Shuai Zhang, Susie Xi Rao, and Ce Zhang. Neural methods for logi333 cal reasoning over knowledge graphs. In The Tenth International Conference on Learning 334 Representations, ICLR 2022, Virtual Event, April 25-29, 2022. OpenReview.net, 2022. URL 335 https://openreview.net/forum?id=tgcAoUVHRIB.

---

> > ### Comment · Reviewer_ktzN · 2023-08-18
> >
> > I have carefully reviewed the comments from other reviewers and the authors' responses. The authors' replies have addressed most of my concerns. Concerning W3, I would appreciate a more comprehensive analysis beyond what is presented in Table 3. Specifically, I am interested in how models constrained exclusively by occurrence can avoid conflicts related to both occurrence and temporal aspects. This curiosity arises from the already provided distribution of contradictory answers in Table 2 (by the way, there are spelling errors for "occurrence" in Tables 2 and 3).
> > My primary recommendation revolves around the structure of the paper. While the introduction of a new task necessitates the provision of background information, I believe that certain content could be drawn from references and the Appendix. What I am emphasizing is a requirement for more intricate model details and insightful experimental analysis in the primary content. Thus, I am inclined to maintain my current score.

---

> > > ### Author Response · Authors · 2023-08-19
> > >
> > > We appreciate your time and effort in reading our rebuttal. To address your concerns, we would like to provide further clarification and analysis. We will address each comment individually.
> > >
> > > Comment: How can models constrained exclusively by occurrence avoid conflicts related to both occurrence and temporal aspects?
> > >
> > > Response: We would like to clarify that our model is not exclusively constrained by occurrences. As depicted in Figure 3, the model is constrained by informational atomics, which may include either occurrence or temporal constraints. These constraints are constructed and filtered using the z3 prover. As explained in the paper (lines 229-240), we create our data using theorem provers for occurrence constraints (True or False) and a linear program solver for treating the currency time as continuous variables (floating point numbers) to detect potential temporal contradictions.
> > >
> > > Our MEQE model captures the constraints within informational atomics by first computing relevance scores and then adding the corresponding constraint information to the query embedding. The effectiveness of this method is demonstrated in Table 3. Our model can capture implicit constraints, and the content in Table 3 shows that it works for both occurrence and temporal constraints.
> > >
> > > Comment: Model details and experimental analysis
> > >
> > > Response: As mentioned earlier, Table 3 shows the improved performance of MEQE across four different backbones on a dataset that includes queries with both occurrence and temporal constraints. To further explain the performance improvement of MEQE, we conducted an ablation study, as detailed in our reply to reviewer YBho.
> > >
> > > Here are the key findings from the ablation study:
> > >
> > > The relevance score effectively identifies corresponding constraints. The feed-forward layer is essential for adjusting the direction of memory contents to incorporate into the query embedding. Removing the FFN layer significantly reduces performance.
> > >
> > > We hope our additional explanations and ablation study results provide further insights into Table 3 and adequately address your concerns.
> > >
> > >
> > > | Models |  | Occurrence Constraints |  |  | Temporal Constraints |  |  | Average |  |  |  |
> > > |---|:---:|:---:|:---:|:---:|:---:|:---:|:---:|:---:|:---:|---|---|
> > > |  | Hit@1 | Hit@3 | MRR | Hit@1 | Hit@3 | MRR | Hit@1 | Hit@3 | MRR |  |  |
> > > | GQE | 8.92 | 14.21 | 13.09 | 9.09 | 14.03 | 12.94 | 9.12 | 14.12 | 13.02 |  |  |
> > > | + CMQE | 10.20 | 15.54 | 14.31 | 10.70 | 15.67 | 14.50 | 10.45 | 15.60 | 14.41 |  |  |
> > > | + CMQE - Constraints | 8.29 | 12.87 | 11.62 | 8.80 | 13.02 | 12.17 | 8.54 | 12.95 | 11.90 |  |  |
> > > | + CMQE - FFN | 0.67 | 1.17 | 1.13 | 0.74 | 1.23 | 1.12 | 0.70 | 1.19 | 1.08 |  |  |
> > > | Q2P | 14.14 | 19.97 | 18.84 | 14.48 | 19.69 | 18.68 | 14.31 | 19.83 | 18.76 |  |  |
> > > | + CMQE | 15.15 | 20.67 | 19.38 | 16.06 | 20.82 | 19.74 | 15.61 | 20.74 | 19.56 |  |  |
> > > | + CMQE - Constraints | 14.16 | 20.00 | 18.86 | 14.72 | 19.92 | 18.79 | 14.44 | 19.96 | 18.82 |  |  |
> > > | + CMQE - FFN | 12.77 | 16.63 | 15.89 | 12.74 | 16.83 | 14.75 | 12.76 | 16.73 | 15.32 |  |  |
> > > | Nerual MLP | 13.03 | 19.21 | 17.75 | 13.45 | 19.06 | 17.68 | 13.24 | 19.14 | 17.71 |  |  |
> > > | + CMQE | 15.26 | 20.69 | 19.32 | 15.91 | 20.63 | 19.47 | 15.58 | 20.66 | 19.40 |  |  |
> > > | + CMQE - Constraints | 13.33 | 19.15 | 17.94 | 13.49 | 19.18 | 14.48 | 13.41 | 19.16 | 18.08 |  |  |
> > > |  + CMQE - FFN | 10.35 | 14.67 | 13.71 | 10.94 | 14.67 | 12.74 | 10.64 | 14.67 | 14.53 |  |  |
> > > | FuzzQE | 11.68 | 18.64 | 17.07 | 11.68 | 17.97 | 16.53 | 11.68 | 18.31 | 16.80 |  |  |
> > > | + CMQE | 14.76 | 21.12 | 19.45 | 15.31 | 21.01 | 19.49 | 15.03 | 21.06 | 19.47 |  |  |
> > > | + CMQE - Constraints | 12.69 | 19.92 | 17.68 | 13.53 | 18.25 | 17.91 | 13.11 | 19.08 | 17.80 |  |  |
> > > | + CMQE - FFN | 9.81 | 15.26 | 14.46 | 10.17 | 15.37 | 14.87 | 9.99 | 15.31 | 14.66 |  |  |
> > >
> > >
> > > Re: Restructuring the paper.
> > >
> > > Thank you for your suggestions on restructuring our paper. As we are unable to modify the paper directly this year, we will implement your suggestions by drawing the details of model definitions and parameterizations of backbone models from the reference and appendix and incorporating them into the primary content. Additionally, we will move the detailed logical definitions and examples to the Appendix.

---

### Official Review · Reviewer_DAya · 2023-07-07

**Soundness:** 1 poor
**Presentation:** 2 fair
**Contribution:** 1 poor
**Rating:** 3
**Confidence:** 4

**Summary:**

The paper proposes a new task of complex event knowledge graph completion, curates the datasets from existing event knowledge graph, and designs a solution. However, the tasks motivation is not very clear regarding the formal logic form as each event is still in the form of natural language, especially considering the event chains are likely break down along with increasing reasoning paths. The authors need to further discuss the different and importance of the proposed task compared with commonsense reasoning or commonsense knowledge graph completion.

Weaknesses

1. The paper is hard to follow and needs further polish. For example,
    1. line 53-60, what ‘s the difference between entity-centric and event centric KGC? The given example is not convincing and seems artificial. If the event does not occur, it may not be necessary to included in the KG. So, the difference lies on the Open-world assumption or closed world assumption, not entity- and event-centric.
    2. What is the motivation to formulate the commonsense knowledge as the formal logic form? For commonsense knowledge, it has a great chance to hold true with probability. That is, along with the increasing paths, the reasoning chains are more likely false.
2. The proposed task looks similar with commonsense reasoning. So, it is necessary to discuss and compare with the datasets and baseline methods for commonsense question answering or commonsense knowledge graph completion.
3. What is the quality of the curated benchmark?

**Strengths:**

see summary

**Weaknesses:**

see summary

**Questions:**

see summary

**Limitations:**

see summary

---

> ### Author Rebuttal · Authors · 2023-08-09
>
> |  | Commonsense KG |  | Eventuality KG |
> |---|---|---|---|
> | KG  | Atomic | ConceptNet | ASER |
> | Node | Event-Centric (mostly) | Entity-Centric (mostly) | Eventuality |
> | Edges | If A then B;  | Entity relations  | If A then B;  |
> |  | A because B; | … | A because B; |
> |  | A as a result B;  |  | A as a result B;  |
> |  | A before B;  |  | A before B;  |
> |  | A after B. |  | A after B. |
> |  |  |  | Narrative Relations:  |
> |  |  |  | **Although A, B;** |
> |  |  |  | **A but B;** |
> |  |  |  | **A and B;** |
> |  |  |  | **A for example B;** |
> |  |  |  | **A in other words B;** |
> |  |  |  | **A or B;** |
> |  |  |  | **Instead of A, B;** |
> |  |  |  | **A except B;** |
>
>
> First of all, we would like to clarify that **not all relations between events/eventualities are commonsense relations**.  As we have explained in the section 2 about  the scope of relations we adopt in this paper is discourse relations, which include four general types: **temporal** (before/after), **contingency** (because/result), **comparison** (but/although), and **expansions** (and/or/except/instead). The commonsense relations, on the other hand, mainly focus on the first two types of relations: **contingency** and **temporal**. The occurrence constraints discussed in this paper primarily exists in the **expansion** type which do not appear in commonsense KG but exists in the KG.
>
> Based on this clarification, we will address your concerns:
>
> Re W1.1:
>
> As shown in line 33-34 and line 44-45, the major difference is that the vertices of entity-centric KG are entities, and the vertices of event-centric KGs are events. On simple way to distinguish events from entities is that, the events include verbs that that can be associated with ture/false values to indicate whether that events happens in a specific situation, but the entities are all nouns.
>
> Meanwhile, the open-world assumption means the missing edges in a knowledge graph are unknown instead of false. Regardless of what kind of KG is, this assumption is always adopted in almost every task for knowledge graph, like KG completion, complex query answering, and rule mining.
>
> Re W1.2
>
> Yes, as you have said, commonsense knowledge is something that mostly happens in a situation. But the level of commonsense modeling is not down to the level of our modeling of events, in which we care about under a specific situation, the occurrence of events.  This is why we need logical reasoning in our task.
>
>
> Re: W2
>
> Difference between Commonsense Question Answering and Complex Eventuality Query Answering (CEQA).
>
> CommonsenseQA is a benchmark dataset for NLP models to test the ability to reason with commonsense knowledge, and it's a multiple choice problem. The commonsense QA problems do not necessarily include events and the relations between events. Here is an example:
>
> * Where can I stand on a river to see water falling without getting wet?
> * A. waterfall,  B. bridge,  C. valley, D. stream, E. bottom
>
>
> Commonsense KG completion is the task of given head and relation, predicting the tail node. The evaluation is a ranking task.
>
> For ConceptNet:
> * Question: (bacteria, causes, V)
> * Answers: [tooth decay, infection in cut]
>
> For Atomic:
> * Question: (X repels Y's attack, Xattribute, ?)
> * Answers: [X is strong, X is skilled, X is brave]
>
> Our task differs from traditional question answering tasks because our task encompasses a broad range of relationships, including non-common sense relations as seen in Treebank 2.0, which provides numerous additional relations. Some complex queries include relations that only exist at eventuality level, such as co-ocurence, conjunction, contradiction, cannot be effectively simulated by commonsense question answering. As a result, our focus lies in complex query answering, where queries primarily revolve around intricate connections between eventualities. Unlike existing common sense knowledge graphs (CSKGs) that can only replicate relations involving two events in a triple, our task involves far more than two events in a single query-answer pair. This poses a challenge in formulating our task as either a knowledge graph completion (KGC) or question-answering (QA) task, as such formulations would necessitate discarding most query constraints, diminishing complexity, and transforming it into a simpler query answering task. Although commonsense knowledge may come into play when answering our queries, it is not as prevalent as in other tasks. Instead of relying on human-level commonsense reasoning, our task also doesn’t rely on the semantic information of the query, but rather rely on learning graph structures to perform query answering and reasoning, utilizing the structure of the graph itself instead of natural language. Thus, we believe that our task is significantly different and methods solving those tasks cannot be simply mitigated.
>
> Re: W3
>
> Our benchmark dataset is derived directly from ASER without any complex transformations. ASER has been demonstrated to possess exceptional eventuality and discourse extraction quality, with an accuracy of over 90%. This means that the extracted information is highly accurate and representative of the original semantic meanings. As a result, we are confident in the reliability of our benchmark's query-answer pairs.
>
> We used the Amazon Mechanical Turk platform to conduct human annotations and assess the plausibility of each query-answer pair. The workers received thorough training, including detailed instructions and qualification rounds to ensure their accuracy level was above 90%. After annotating 1200 query-answer pairs, we calculated the statistics and found that 86.5% of the answers were considered plausible. These results align with our expectations and provide validation for our hypothesis.
>
> We assure you that we will carefully consider these discussions and incorporate them into our paper's final version. Additionally, we plan to conduct case studies to further demonstrate the quality of our benchmark.

---

### Decision · Program_Chairs · 2023-09-21

**Decision:**

Accept (poster)

**Comment:**

I support accepting this paper. All reviewers besides Reviewer DAya support accepting the paper.

I read the author response and think the authors have sufficiently answered Reviewer DAya's concerns (the reviewer did not reply to the response). In particular,

-As the authors state, their paper is substantially different from common sense reasoning:  "Some complex queries include relations that only exist at eventuality level, such as co-ocurence, conjunction, contradiction, cannot be effectively simulated by commonsense question answering. "

-The authors addressed the concern about the quality of the benchmark by discussing the quality of ASER as well as the results of an Amazon Mechnical Turk evaluation.